# JointAVBench: A Benchmark for Joint Audio-Visual Reasoning Evaluation

**Jianghan Chao[1, 3], Jianzhang Gao[1], Wenhui Tan[1], Yuchong Sun[1], Ruihua Song[1],*, Liyun Ru[2]**
[1]Gaoling School of Artificial Intelligence, Renmin University of China
[2]Baichuan Inc
[3]Zhongguancun Academy
{chaojh, rsong}@ruc.edu.cn

## Abstract

Understanding videos inherently requires reasoning over both visual and auditory information. To properly evaluate Omni-Large Language Models (Omni-LLMs), which are capable of processing multi-modal information, including vision and audio, an effective benchmark must comprehensively cover three key aspects: (1) multi-modal dependency (i.e., questions that cannot be answered using vision or audio alone), (2) diverse audio information types (e.g., speech, sound events), and (3) varying scene spans. However, existing datasets fall short in one or more of these dimensions, limiting strict and comprehensive evaluation. To address this gap, we introduce JointAVBench, a novel benchmark with strict audio-video correlation, spanning five cognitive dimensions, four audio information types (speech, sound events, music, vocal traits), and three scene spans (single-, cross-, and full-scene). Given the high cost of manual annotation, we propose an automated pipeline that leverages state-of-the-art vision-LLMs, audio-LLMs, and general-purpose LLMs to synthesize questions and answers that strictly require joint audio-visual understanding. We evaluate leading vision-only, audio-only, and Omni-LLMs on our dataset. Results show that even the best-performing Omni-LLM achieves an average accuracy of only 65.3%, outperforming uni-modal baselines but revealing substantial room for improvement, especially in cross-scene reasoning.
**Project page**: `https://roverx12345.github.io/jointavbench_webpage/`

## 1 Introduction

Humans can understand videos and the real world by seamlessly perceiving and integrating both visual and auditory information across different scenes, where diverse audio signals (*e.g.* speech, sound, music, or even vocal traits) are used to complement the visual scene in analyses. As illustrated in Figure 1(a), for a multi-scene video understanding task, such as determining the order across a visual object in scene 3 with the dialogue in scene 1 and scene 23, requires complex joint audio-visual reasoning. This process involves recognizing visual and auditory cues, correlating them across distinct temporal and spatial contexts, and reasoning with the acquired relations. Toward the goal of artificial general intelligence, equipping multimodal large language models (MLLMs) with such joint audio-visual reasoning ability is paramount.

While newly developed Omni-LLMs (Team et al., 2023; Bai et al., 2025; Han et al., 2024; Wu et al., 2024a; Su et al., 2023) aim to process both audio and visual inputs jointly, progress is hindered by the lack of a comprehensive benchmark dedicated to evaluating this crucial capability. Existing benchmarks exhibit several limitations: some lack strict audio-visual correlation controls (Hong et al., 2025; Geng et al., 2024), others primarily focus on static images or simple videos (Li et al., 2024c; Gong et al., 2024), and mostly cover only a limited range of audio types (Yang et al., 2025). Furthermore, nearly all existing benchmarks neglect the complexities of multi-scene reasoning, which is a core component of human cognition.

---

*Corresponding author.

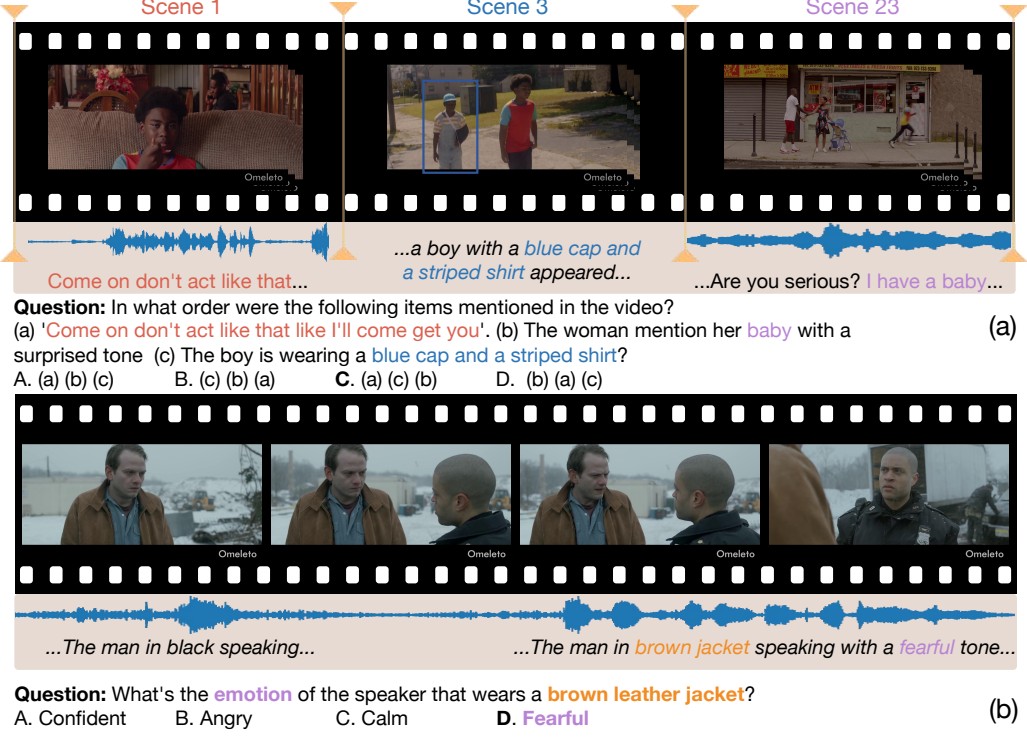

Figure 1: **Examples of JointAVBench.** (a) asks a cross-scene plot-related question that needs the visual information in Scene 3 and the speech information in Scene 1 and Scene 23 to reason the right answer. (b) asks a single-scene emotion-related question that needs the visual information of the speaker and his vocal traits to answer.

To address this critical gap, we introduce JointAVBench, the first comprehensive benchmark for evaluating Omni-LLMs' joint audio-visual reasoning capabilities. Our benchmark features a systematic taxonomy covering five cognitive dimensions (e.g., temporal, plot, and long-form reasoning), four audio signal types (vocal traits, music, speech, and sound event), and three distinct scene spans (single-, cross-, and full-scene). These features enable us to construct 15 challenging tasks with strict audio-visual correlations, providing a unified and rigorous evaluation framework. For example, the task in Figure 1 (b) tests Speaker Emotion Recognition (SER), i.e., a single-scene, vocal traits involved, and emotion-related task.

To overcome the immense cost of manual annotation, we propose a semi-automated pipeline to generate high-quality question-answer (QA) pairs. This three-stage process first generates detailed multimodal captions, then synthesizes questions that strictly require joint audio-visual reasoning, and finally performs a rigorous quality assurance step to ensure data fidelity. We then use human labor to filter out unqualified data. This approach enables us to construct a high-quality multi-choice benchmark of 2,853 samples that are designed to probe complex reasoning abilities.

We conduct extensive experiments on JointAVBench to evaluate three types of MLLMs: Omni-LLMs, Video-LLMs, and Audio-LLMs. Our results demonstrate that current Omni-LLMs, such as Qwen-omni and Gemini-2.5 flash, significantly outperform their single-modal counterparts. However, our analysis also reveals that these models exhibit uneven capabilities across different audio types and suffer from a substantial performance degradation with increasing scene complexity. Our comprehensive assessment highlights critical limitations in current models' audio-visual reasoning capacities, posing the potential for future improvement.

In summary, our contributions can be summarized as follows:

Table 1: Comparison between our benchmark and previous ones. **Anno.:** the construction method, where A for automatic pipeline, A+M for pipeline involving manual inspection, and M for manual pipeline. **Modality:** the modality involved. V for video, I for image, and A for audio. **Aud. Type:** number of different audio signal types included in the dataset or benchmark. **AV Corr. Ratio:** the ratio of true audio-visual correlated questions, discussed in detail in Appendix C.1. † This metric is evaluated only on the three MCQ audio-visual tasks of AVUT (with two additional audio-visual open-ended tasks yet to be released).

| Benchmark/Dataset | Avg. Duration | #QA | Anno. Method | Modality | #Tasks | #Audio Types | AV Corr. Ratio |
|---|---|---|---|---|---|---|---|
| ***Video Benchmarks/Datasets*** | | | | | | | |
| EgoSchema (Mangalam et al., 2023) | 180s | 5,063 | A+M | V | 1 | - | 0 |
| Video-MME (Fu et al., 2025a) | 1,017.9s | 2,700 | M | V | 12 | - | 0 |
| MVBench (Li et al., 2024b) | 16.0s | 4,000 | A | V | 20 | - | 0 |
| LVBench (Wang et al., 2024b) | 4,101s | 1,549 | M | V | 6 | - | 0 |
| MMBench-Video (Fang et al., 2024) | 165.4s | 1,998 | M | V | 26 | - | 0 |
| ***Audiovisual Benchmarks/Datasets*** | | | | | | | |
| Music-AVQA (Li et al., 2022a) | 60s | 45,867 | M | V&A | 9 | 1 | 56.7% |
| OmniBench (Li et al., 2024c) | - | 1,142 | M | I&A | 8 | 3 | 80.4% |
| AV-Odyssey (Gong et al., 2024) | - | 4,555 | M | V/I&A | 26 | 3 | 99.0% |
| LongVALE (Geng et al., 2024) | 235s | - | A+M | V&A | 3 | 3 | 76.2% |
| AVUT (Yang et al., 2025) | 67.8s | 13,774 | A+M | V&A | 8 | 2 | 77.8%† |
| WorldSense (Hong et al., 2025) | 141.1s | 3,172 | M | V&A | 26 | 3 | 62.9% |
| **JointAVBench (ours)** | 97.2s | 2,853 | A+M | V&A | 15 | 4 | 93.5% |

- We introduce JointAVBench, the first-ever comprehensive benchmark to evaluate joint audio-visual reasoning capability across five cognitive dimensions, four audio types, and three scene complexities.

- We propose a novel three-stage semi-automated pipeline for generating high-quality QA pairs with strict audio-visual correlations while reducing annotation difficulties and costs.

- We provide a comprehensive evaluation of current MLLMs on JointAVBench, demonstrating their limitations and highlighting the importance of developing truly integrated audio-visual reasoning Omni-LLMs.

## 2 RELATED WORKS

### 2.1 MULTIMODAL LARGE LANGUAGE MODELS

The rise of Large Language Models (LLMs) has spurred interest in extending their capabilities beyond text to multimodal inputs (Bi et al., 2024; Achiam et al., 2023; Radford et al., 2018). Early efforts, such as (Radford et al., 2021; Hurst et al., 2024; Li et al., 2022b; 2023b), demonstrate effective fusion of visual and textual modalities for cross-modal understanding. Subsequent studies (Chu et al., 2023; Radford et al., 2023) expand this paradigm to incorporate audio-text integration and achieve significant improvements. Later advances in hardware and memory optimization enable video-text modeling in MLLMs (Team et al., 2023; Gao et al., 2017; Geng et al., 2022; Huang et al., 2024), spanning across various domains and achieving progress such as long video understanding (Wang et al., 2024a; Yuan et al., 2025; Chen et al., 2024a) and movie understanding (He et al., 2024; Song et al., 2024). Recent works (Chowdhury et al., 2025; Shu et al., 2023; Cheng et al., 2024; Tang et al., 2025b; Fu et al., 2024; 2025b; Lu et al., 2022; 2024; Su et al., 2023; Wu et al., 2024a; Han et al., 2024) focus on achieving human-like audio-visual joint reasoning ability by interleaving audio, video, and text. This requires datasets to contain QAs with strict audio-visual correlations. To facilitate the development of Omni-LLMs, we present JointAVBench to evaluate the models' audiovisual joint reasoning ability with questions that are fully audio-visual correlated.

### 2.2 AUDIO-VISUAL BENCHMARKS

With the development of MLLMs, various benchmarks have been constructed for the evaluation of MLLMs' comprehensive abilities (Wu et al., 2024b; Li et al., 2024a; 2023a; Yue et al., 2024; Zhang et al., 2024a; Sakshi et al., 2024; Liu et al., 2024; Li et al., 2024b). Early datasets or benchmarks, such

Table 2: Task categories of our designed taxonomy. In audio signal type, we use SPE for speech, VOT for vocal traits, SEV for sound event, and MUS for music.

| Scene Type | Cognitive Dimension | Audio Signal Type | Task Name | Task Code |
|---|---|---|---|---|
| Single | Temporal | SPE | Speech-based Timepoint Localization | STL |
| | | SPE | Vision-Speech Sequence Recognition | VSSR |
| | Spatial | VOT | Speaker Spatial Localization | SPL |
| | | SEV | Sounding Object Grounding | SOOG |
| | | SEV | Sound Event Recognition | SOER |
| | Emotion | VOT | Speaker Emotion Recognition | SPER |
| | | MUS | Musical Tone Inference | MPTI |
| Multiple | Long-form | SPE | Cross-scene Association | CSA |
| | Plot | SPE, VOT | Multi-plot Ordering | MPO |
| | | SPE | Plot Development Prediction | PDP |
| | | SEV, MUS | Audio Function Analysis | AFA |
| | Temporal | SPE | Plot Temporal Grounding | PTG |
| Full | Long-form | SPE, VOT, SEV, MUS | Audio-Visual Detail Memory | AVDM |
| | Emotion | MUS | Musical Emotion Shift Inference | MESI |
| | Plot | SPE | Character Relationship Inference | CRI |

as AVQA (Yang et al., 2022), Music-AVQA (Li et al., 2022a), and AVInstruct (Ye et al., 2024), only focus on certain types of audio signals and lack strict audio-visual correlation. Subsequent works such as Omni-bench (Li et al., 2024c) and AV-Odyssey (Gong et al., 2024) consist primarily of only image and audio, lacking the evaluation of videos. The recent WorldSense (Hong et al., 2025) has delved into the problem. However, it lacks strict audio-visual correlation and emphasizes the evaluation of visual tasks. These datasets cannot capture the complex and interleaved auditory and scene details in video (such as the details in Figure 1). In contrast, our proposed JointAVBench focuses on the evaluation of diverse audio signal types and multilevel scenes, aiming to conduct a comprehensive and systematic assessment of current MLLMs for joint audio-visual understanding.

## 3 JOINTAVBENCH

We propose JointAVBench, a benchmark for evaluating Omni-LLMs' joint audio-visual reasoning ability. This section will first detail the benchmark's core requirements, then the carefully designed data generation pipeline, including (i) omni-caption generation, (ii) QA pair creation, and (iii) rigorous quality control, with statistics provided at the end of this section.

### 3.1 BENCHMARK REQUIREMENTS

The benchmark construction adheres to three fundamental requirements, ensuring comprehensive evaluation of joint audio-visual reasoning capabilities.

**Strict Audio-Visual Correlation.** We design a hierarchical taxonomy comprising 15 tasks (detailed in Table 2), ensuring that each task requires the integration of both visual and audio information to generate answers.

**High-quality Video Source.** Movie scenes are a natural source of extensive and diverse multimodal data. For our benchmark, we leverage the Short-Films 20K (SF20K) (Ghermi et al., 2024) dataset, which comprises 1,072 professionally produced movies rich in narrative and balanced audiovisual features. We then remove unavailable or grayscale videos, retaining 1,046 films that can be used to construct our benchmark.

**Multi-dimensional Task Taxonomy.** To ensure a comprehensive and fine-grained evaluation, we categorize our tasks along three key dimensions:

- **Cognitive Dimension:** Derived from a systematic analysis of previous studies (Fu et al., 2025a; Hong et al., 2025), this dimension assesses core cognitive abilities essential for video understanding. We define 5 types of cognitive dimensions: temporal, spatial, emotional, plot, and long-form.

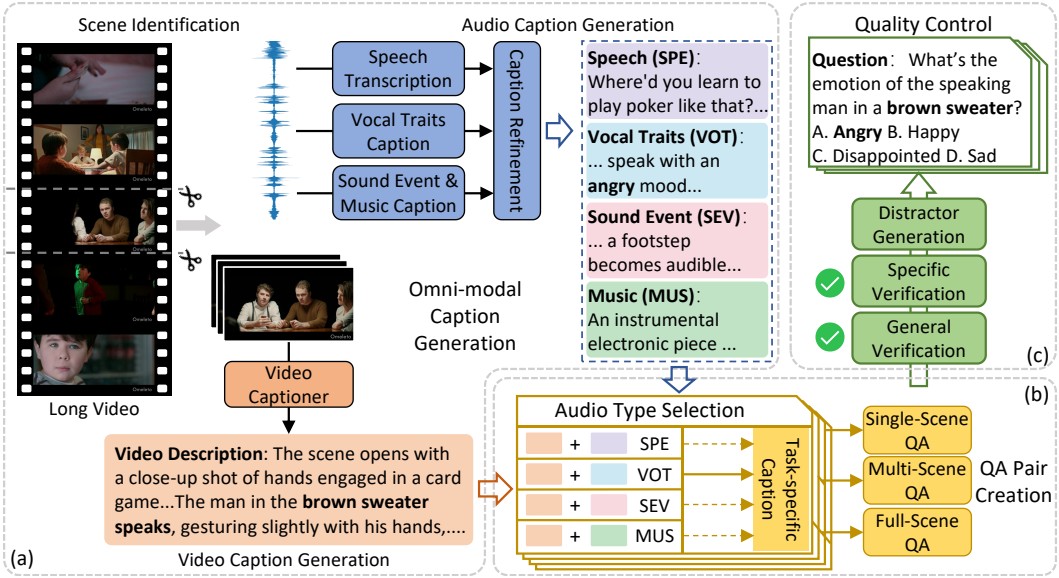

Figure 2: **Pipeline for JointAVBench.** Our construction pipeline is three-fold: (a) Omni-modal caption generation, (b) QA pair creation, and (c) Quality control.

- **Audio Types:** This dimension enables a comprehensive evaluation of audio understanding capabilities across all audio signal types. We divide audio into four types of signals: speech, vocal traits, sound event, and music.

- **Scene Complexity:** This dimension evaluates model performance across videos with varying temporal characteristics, using different scene types to quantify temporal information. We define three types of scene complexity: single-scene, multi-scene, and full-scene.

## 3.2 BENCHMARK CONSTRUCTION

Our dataset construction pipeline is illustrated in Figure 2, which adopts a semi-automated process capable of handling diverse modality characteristics. More details can be found in the appendix.

### 3.2.1 STAGE 1: OMNI-MODAL CAPTION GENERATION

**Scene Identification.** We first split the video into scenes with semantic consistency. Specifically, we follow the procedure in Panda-70m (Chen et al., 2024b) to divide long videos into distinct scenes with PySceneDetect[1] and then merge scenes with high semantic similarity, ensuring in-scene consistency. These segmented scenes retain considerable length, enabling us to capture richer contextual information within each scene.

**Video Caption Generation.** After scene identification, we directly generate visual descriptions for all segmented scenes, ensuring that static features (*e.g.* in-scene objects and characters) and dynamic features (*e.g.* transitions between shots and movements of characters) are well captured.

**Audio Caption Generation.** To ensure the diversity of audio types, we follow the requirements to generate captions for each audio type as shown in Figure 2. Notably, we observe that existing audio models have limitations in distinguishing between sound event and music, and therefore generate their captions simultaneously. Subsequently, we refine the audio captions by addressing the hallucination in the caption and separating the sound event caption and music caption using different LLM judges.

---

[1]https://www.scenedetect.com/

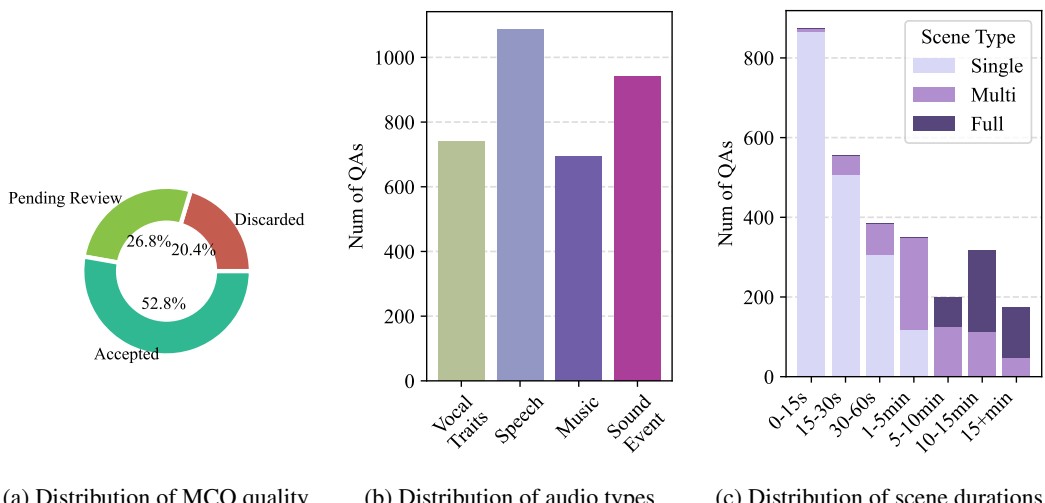

(a) Distribution of MCQ quality     (b) Distribution of audio types     (c) Distribution of scene durations

Figure 3: Statistics of JointAVBench.

### 3.2.2 STAGE 2: QA PAIR CREATION

To create QA pairs with strict audiovisual correlation, we design various question templates for tasks that LLM cannot easily understand (temporal, plot tasks that require complex audio-visual relation), while leaving other tasks to be curated by LLM to ensure question diversity (general tasks such as Character Relationship Inference). Additionally, when providing cross-modal descriptions, we strictly adhere to each task's modality and scene requirements by inputting only the required modality descriptions from the designated scenes. For example, when generating data for the task Speaker Spatial Localization, we provide video captions along with vocal traits descriptions from only one scene. This procedure can eliminate possible interference from extraneous modalities and scenes.

### 3.2.3 STAGE 3: QUALITY CONTROL

We implement a multi-stage quality control process to address issues identified in the collected 9,109 QA pairs, such as mismatched question-answer pairs and redundant information. This process employs a general-to-specific verification strategy, where we guide models to use a chain-of-thought approach for step-by-step data filtering.

**General Verification.** We validate all QA pairs to ensure they meet fundamental standards. This includes a **Modality Check** to confirm that each QA pair necessitates both audio and video information, and a **Logic Check** to verify that answers are directly derivable from the question's context. For instance, a question like *"What is the emotion of the adult male speaker?"* will be discarded if the audio contains only one male speaker, as the answer can be inferred from a single modality.

**Specific Verification.** This stage focuses on task-specific validations. We design the following three specific checks based on QA's task: 1) **Sequence Check** to ensure the correct element order for sequence-based tasks; 2) **Ambiguity Check** to filter out overly generic QA pairs for complex reasoning tasks (e.g., *"What makes the door closing sound?"* with *"door closing"* as the answer); 3) **Audio Signal Type Check** to confirm that the required auditory information cannot be deduced from visual information for sound event and music.

**Distractor Generation.** For each verified QA pair, we craft three plausible but incorrect distractors to create challenging multiple-choice questions. These distractors incorporate diverse misdirections, such as replacing the sound source or confusing details.

### 3.3 HUMAN VERIFICATION

From the automated three-stage generation process, we obtain 3,974 MCQs. To ensure their quality and factual accuracy, we conducted a rigorous human verification process, and the results are

Table 3: Evaluation results of three types of mainstream MLLMs. We evaluate the performance of Omni-LLMs, Video-LLMs and Audio-LLMs on JointAVBench to provide a comprehensive analysis. † For neatness, we use short names of models, where v-SALMONN represents video-SALMONN series.

| Model† | Size | STL | SPL | SOOG | SOER | SPER | MPTI | VSSR | CSA | MPO | PTG | AFA | PDP | AVDM | MESI | CRI | **Avg** |
|---|---|---|---|---|---|---|---|---|---|---|---|---|---|---|---|---|---|
| | | | | | | | *Omni-LLMs* | | | | | | | | | | |
| Gemini2.5-Pro | - | **77.7** | **66.7** | 63.6 | 71.1 | **40.2** | 69.2 | **79.5** | 47.9 | **67.6** | **62.1** | 64.7 | **48.4** | **76.6** | 67.9 | 83.1 | **65.3** |
| Qwen3-Omni | 30B | 68.8 | 59.6 | **73.1** | **77.5** | 39.9 | **75.1** | 76.8 | 45.0 | 57.7 | 32.9 | 50.3 | 47.3 | 67.0 | 72.9 | **85.0** | 63.6 |
| Gemini2.5-Flash | - | 65.2 | 51.1 | 56.9 | 67.5 | 27.6 | 64.5 | 66.5 | 39.7 | 55.3 | 59.3 | 62.3 | 44.1 | 71.7 | 69.2 | 78.5 | 58.0 |
| Qwen2.5-Omni | 7B | 67.8 | 35.3 | 59.5 | 73.5 | 35.2 | 65.6 | 76.3 | 48.8 | 40.4 | 21.5 | **68.2** | 47.3 | 49.1 | 71.4 | 67.3 | 56.5 |
| VideoLLaMA2 | 7B | 20.0 | 42.4 | 53.0 | 67.0 | 33.9 | 50.0 | 47.6 | 24.0 | 33.3 | 30.2 | 58.5 | 35.5 | 39.8 | 63.2 | 59.4 | 46.8 |
| v-SALMONN-2+ | 7B | 37.4 | 25.4 | 57.4 | 58.5 | 21.6 | 67.4 | 52.2 | 27.5 | 42.3 | 22.8 | 52.8 | 43.0 | 39.0 | 72.2 | 59.1 | 46.7 |
| OneLLM | 7B | 29.6 | 37.9 | 44.3 | 36.9 | 26.2 | 32.6 | 33.5 | **51.2** | 29.8 | 32.2 | 44.3 | 41.9 | 34.5 | 35.3 | 48.5 | 36.9 |
| v-SALMONN-o1 | 7B | 28.7 | 30.5 | 34.1 | 42.3 | 17.5 | 41.8 | 31.1 | 25.6 | 18.0 | 36.9 | 52.0 | 30.1 | 35.5 | 66.2 | 55.2 | 36.4 |
| v-SALMONN | 7B | 53.0 | 23.3 | 37.5 | 51.3 | 20.7 | 35.1 | 33.0 | 33.1 | 32.7 | 25.4 | 49.4 | 24.7 | 25.3 | 37.9 | 48.4 | 35.8 |
| AVicuna | 7B | 31.0 | 30.2 | 33.4 | 38.2 | 17.9 | 31.9 | 22.5 | 27.5 | 23.2 | 30.1 | 41.2 | 33.3 | 29.5 | 31.8 | 44.3 | 31.0 |
| | | | | | | | *Video-LLMs* | | | | | | | | | | |
| InternVL-2.5 | 8B | 26.1 | 40.2 | 60.1 | 71.1 | 31.9 | 62.7 | 52.2 | 40.8 | 44.2 | 27.5 | 59.7 | 40.9 | 50.0 | 67.7 | 67.1 | 51.7 |
| VideoLLaMA3 | 7B | 40.0 | 44.6 | 58.4 | 55.8 | 25.2 | 63.8 | 49.6 | 33.9 | 45.2 | 33.6 | 59.7 | 41.9 | 50.9 | 73.7 | 63.0 | 50.0 |
| Qwen2.5-VL | 7B | 33.0 | 38.4 | 55.3 | 61.0 | 27.9 | 58.0 | 47.6 | 29.2 | 41.3 | 32.2 | 60.8 | 36.6 | 40.7 | 65.2 | 61.6 | 47.7 |
| LLaVA-Video | 7B | 34.8 | 35.7 | 49.3 | 63.3 | 15.3 | 63.8 | 53.5 | 28.9 | 46.2 | 29.5 | 49.4 | 36.6 | 46.4 | **78.2** | 61.8 | 47.1 |
| GPT-4o | - | 31.3 | 39.3 | 55.4 | 70.3 | 18.8 | 55.8 | 24.3 | 39.7 | 17.3 | 14.8 | 53.5 | 43.0 | 51.8 | 57.1 | 72.1 | 45.0 |
| | | | | | | | *Audio-LLMs* | | | | | | | | | | |
| Kimi-Audio | 7B | 51.3 | 26.8 | 48.0 | 61.1 | 36.9 | 47.8 | 37.0 | 40.5 | 32.0 | 26.2 | 61.4 | 36.6 | 40.2 | 56.1 | 66.5 | 45.6 |
| Qwen2-Audio | 7B | 56.8 | 23.8 | 38.0 | 53.1 | 35.0 | 40.4 | 34.3 | 31.1 | 38.2 | 27.6 | 55.0 | 27.4 | 30.4 | 46.6 | 54.8 | 39.5 |

illustrated in Figure 3a. Specifically, a team of human annotators rated QAs based on four key criteria: (i) answer correctness, (ii) information correctness, (iii) audio-visual dependency, and (iv) question difficulty. Based on these ratings, we categorize these data into three subsets: (1) Accepted: QAs that pass the answer correctness check and score highly on all other criteria, which are directly retained in the final dataset; (2) Pending Review: QAs that pass the answer correctness check but receive lower ratings on one or more additional criteria, which are subject to further selection according to their ratings; and (3) Discarded: QAs that fail the answer correctness check and are removed from the dataset.

In total, we retained 2,853 QAs, achieving a data retention rate of 71.8%, which demonstrates that our automatic pipeline is highly effective at generating data of sufficient quality. After this sample-level human verification, we further conduct a post-audit answer-label refinement step to reduce residual label inconsistencies. This post-audit step only refines answer labels within the retained benchmark and does not remove samples or change the benchmark size.

### 3.4 BENCHMARK STATISTICS

JointAVBench consists of 2,853 high-quality, manually verified MCQs spanning all scene levels and audio types, with an average duration of 97.2s (Table 1). A detailed statistical analysis of the benchmark is presented in Figure 3. The number of QA pairs is balanced across diverse audio signal types (Figure 3b), showcasing our benchmark's comprehensive coverage. Moreover, our dataset spans a wide range of video durations (Figure 3c), with single-scene, multi-scene, and full-scene tasks mainly comprising videos of less than 1 min, 1-10 min, and over 10 min, respectively.

## 4 EXPERIMENT

This section first demonstrates a comprehensive evaluation of mainstream MLLMs on our proposed benchmark, and then key factors that influence performance to provide valuable insights for future Omni-LLMs.

## 4.1 EXPERIMENT SETUP

**Models.** Our experiments are conducted on a diverse set of mainstream MLLMs. To comprehensively evaluate their joint audio-visual reasoning capability across different modalities, we categorize them into three groups: (i) Omni-modal LLMs: Qwen3-Omni (Xu et al., 2025b), Qwen2.5-Omni (Xu et al., 2025a), VideoLLaMA2 (Cheng et al., 2024), video-SALMONN-2+ (Tang et al., 2025a), video-SALMONN-o1 (Sun et al., 2025), video-SALMONN (Sun et al., 2024), OneLLM (Han et al., 2024), AVicuna (Tang et al., 2025b), Gemini2.5-Pro (Comanici et al., 2025), and Gemini2.5-Flash (Comanici et al., 2025); (ii) Video-LLMs: Qwen2.5-VL (Bai et al., 2025), LLaVA-Video (Zhang et al., 2024b), Video-LLaMA3 (Zhang et al., 2025), InternVL2.5 (Chen et al., 2024c), and GPT-4o (Hurst et al., 2024); and (iii) Audio-LLMs: Kimi-Audio (Ding et al., 2025) and Qwen2-Audio (Chu et al., 2024).

**Metrics and Experiment Settings.** To achieve evaluation consistency, we follow previous works (Hong et al., 2025; Fu et al., 2025a) and use accuracy as the evaluation metric. For a fair evaluation, we adopt the following protocols for all experiments. For open-source models, we use their official codebase with default configurations, while for closed-source models, we use their official APIs while keeping the configuration by default. To maintain comparability, we select open-source models with comparable 7B parameter sizes and enforce a unified sampling of 32 frames across all models. We also ensure that the text input to all models is limited to the question text, without any additional contextual information. Moreover, due to the deterministic inference setting, results are stable across runs.

## 4.2 RESULTS AND FINDINGS

**Overall Performance.** Our results, summarized in Table 3, reveal that current mainstream MLLMs perform sub-optimally on our benchmark, with the best performing model having an average accuracy of only 65.3%. This suggests a significant gap in their ability to process omni-modal information. Importantly, Omni-LLMs consistently outperform Video-LLMs and Audio-LLMs, highlighting the critical role of native modality integration. For instance, Gemini2.5-Pro achieves the best overall performance, while Qwen2.5-Omni remains competitive with strong open-source video and audio baselines across most tasks.

**Breakdown Findings.** To gain a deeper understanding, we analyze model performance across various task categories and have the following observations.

**1) Models perform unevenly across different audio types, failing on tasks requiring vocal traits and speech.** We find a significant performance gap among different audio types (Figure 4). Models excel at tasks involving sound events and music, likely due to their stronger visual correspondence (where audio often matches visible objects or atmosphere). However, they struggle with more abstract audio, such as speech and vocal traits. This is likely due to a lack of training data focused on vocal traits, as most audio-visual datasets (Li et al., 2022a; Yang et al., 2022) overlook information like emotion and gender, leading to tasks like SPL, SPER, and MPO being the worst-performing overall.

**2) Multi-scene tasks usually yield worse results compared to single-scene tasks, while full-scene tasks often achieve better results.** Figure 5 shows the pronounced impact of scene complexity. Models perform well on single-scene tasks, particularly those requiring speech-based reasoning like STL and VSSR. This is likely because single scenes offer stable visual contexts and limited speech content, simplifying cross-modal correspondence. Conversely, multi-scene tasks requiring speech, such as MPO and PTG, yield worse performance, as they demand more complex processing of diverse scenes and cross-scene connections. Interestingly, while most models struggle with multi-scene tasks, they perform better on full-scene tasks, which focus on global narratives rather than fine-grained details. This highlights that improving models' cross-scene reasoning capabilities is needed.

**3) Omni-models perform worse on emotional and spatial tasks than single-modal models.** As demonstrated in Figure 6, while Omni-LLMs generally perform best in 11 of our 15 tasks, they surprisingly fall behind single-modality models on emotion-based tasks. This suggests that in some cases, single-modality models can better focus on emotion cues without the distraction of additional modalities. Furthermore, Omni-LLMs perform poorly on spatial tasks like SOOG and SOER, even falling behind Video-LLMs. This is likely because models primarily rely on spatial information from video and fail to effectively integrate complementary audio cues. This finding highlights a critical limitation in current models' ability to perform true audio-visual spatial reasoning.

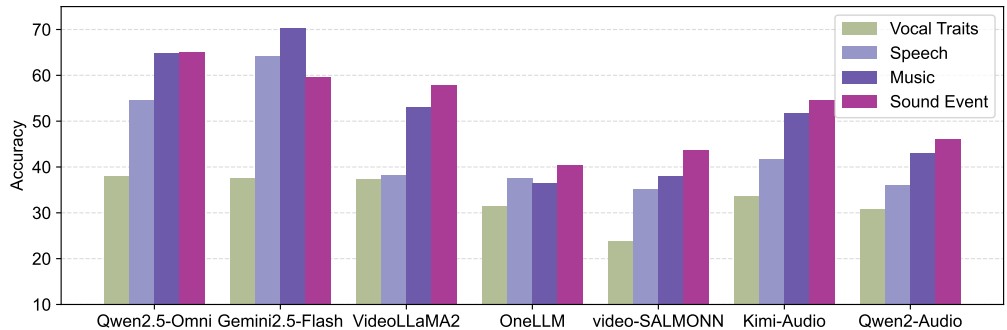

Figure 4: Results on JointAVBench across different audio types.

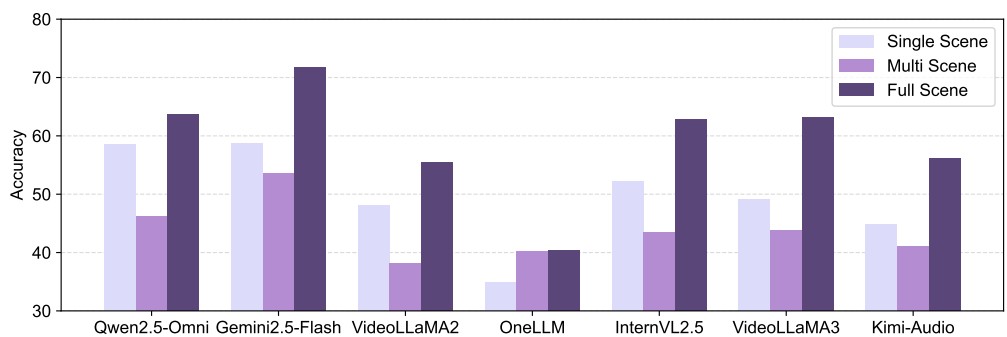

Figure 5: Results on JointAVBench across different scene types.

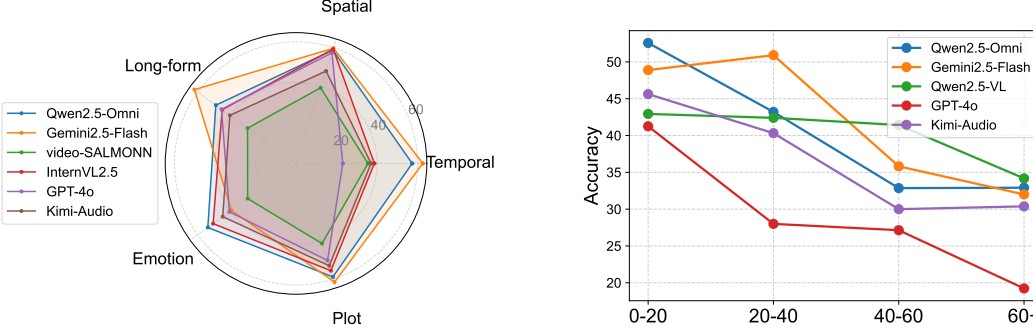

Figure 6: Results on JointAVBench across 5 cognitive dimensions.

Figure 7: Results on JointAVBench with different number of scenes.

**4) Increased scene number leads to models' performance degradation on multi-scene tasks.** Our analysis reveals that increasing the number of scenes adversely affects multi-scene task performance (Figure 7), with accuracy dropping sharply by approximately 20% from 0-20 to over 60 scenes. This underscores the uneven performance of current MLLMs across diverse scenes, highlighting a critical area for improvement.

**5) Omni-modal Models Demonstrate Effective Modality Fusion.** We quantify the effectiveness of joint reasoning by defining $N_o$ as the number of tasks where a model's audio-visual (A+V) performance is higher than the best single-modality score, and $N_u$ as the number of tasks where A+V performance is lower than the worst single-modality score, in Table 4. For most models, $N_o$ outweighs $N_u$, confirming that integrating audio and video generally enhances overall reasoning capability. Furthermore, as a model's overall performance increases, its $N_o$ count rises while its $N_u$ count falls. This pattern indicates that more advanced models are better adept at modality fusion.

Table 4: Evaluation results of open-source omni-LLMs with different modality utilization. † For neatness, we use short names of models, where Qwen2.5 represents Qwen2.5-Omni, ViLLaMA2 represents VideoLLaMA2, and v-SALMONN represents video-SALMONN.

| Model[†] | Modality | $N_o$ | $N_u$ | Avg | STL | SPL | SOOG | SOER | SPER | MPTI | VSSR | CSA | MPO | PTG | AFA | PDP | AVDM | MESI | CRI |
|---|---|---|---|---|---|---|---|---|---|---|---|---|---|---|---|---|---|---|---|---|
| Qwen2.5 | A+V | | | 56.5 | 67.8 | 35.3 | 59.5 | 73.5 | 35.2 | 65.6 | 76.3 | 48.8 | 40.4 | 21.5 | 68.2 | 47.3 | 49.1 | 71.4 | 67.3 |
| | V | 9 | 1 | 49.7 | 36.5 | 33.9 | 56.4 | 66.0 | 25.2 | 60.9 | 49.6 | 40.5 | 41.3 | 23.5 | 65.9 | 38.7 | 49.1 | 72.2 | 65.5 |
| | A | | | 51.8 | 67.0 | 23.9 | 63.7 | 66.2 | 36.5 | 56.2 | 57.0 | 42.1 | 45.2 | 18.1 | 63.1 | 44.1 | 48.2 | 66.2 | 64.6 |
| VidLLaMA2 | A+V | | | 46.8 | 20.0 | 42.4 | 53.0 | 67.0 | 33.9 | 50.0 | 47.6 | 24.0 | 33.3 | 30.2 | 58.5 | 35.5 | 39.8 | 63.2 | 59.4 |
| | V | 5 | 5 | 47.2 | 31.3 | 43.3 | 54.1 | 67.7 | 29.6 | 51.4 | 43.9 | 28.9 | 47.0 | 26.2 | 59.1 | 37.6 | 40.7 | 62.4 | 52.7 |
| | A | | | 40.9 | 24.3 | 33.5 | 54.4 | 60.6 | 20.6 | 36.6 | 31.7 | 36.4 | 33.7 | 29.5 | 60.2 | 30.1 | 31.8 | 50.4 | 53.9 |
| OneLLM | A+V | | | 36.9 | 29.6 | 37.9 | 44.3 | 36.9 | 26.2 | 32.6 | 33.5 | 51.2 | 29.8 | 32.2 | 44.3 | 41.9 | 34.5 | 35.3 | 48.5 |
| | V | 8 | 2 | 32.1 | 25.2 | 35.3 | 36.8 | 27.5 | 17.3 | 26.8 | 30.4 | 31.4 | 32.7 | 31.5 | 39.8 | 51.6 | 33.6 | 34.6 | 50.9 |
| | A | | | 37.2 | 31.3 | 37.1 | 40.9 | 36.4 | 29.6 | 31.5 | 26.8 | 43.0 | 24.3 | 29.5 | 40.9 | 62.4 | 45.8 | 40.6 | 59.4 |
| v-SALMONN | A+V | | | 35.8 | 53.0 | 23.3 | 37.5 | 51.3 | 20.7 | 35.1 | 33.0 | 33.1 | 32.7 | 25.4 | 49.4 | 24.7 | 25.3 | 37.9 | 48.4 |
| | V | 4 | 3 | 34.3 | 20.0 | 25.2 | 44.1 | 47.5 | 17.8 | 34.4 | 27.0 | 29.2 | 42.3 | 25.5 | 47.7 | 40.9 | 30.2 | 37.6 | 41.0 |
| | A | | | 35.7 | 53.0 | 26.2 | 38.3 | 48.9 | 22.4 | 37.1 | 30.4 | 34.7 | 32.7 | 25.0 | 42.6 | 32.3 | 24.8 | 38.9 | 44.2 |

For instance, while early Omni-LLMs (Cheng et al., 2024; Han et al., 2024; Sun et al., 2024) show only marginal gains over single-modality baselines, Qwen2.5-Omni's dual-modality performance significantly surpasses its single-modality results. This confirms that true joint reasoning is a hallmark of mature omni-modal architectures.

## 5 CONCLUSION

In this paper, we propose JointAVBench, a comprehensive benchmark for evaluating joint audio-visual reasoning, distinguished by a hierarchical taxonomy and a high-quality, automated generation pipeline. Each question in JointAVBench is meticulously designed to necessitate the integrated understanding of both visual and a specific type of audio input. We further ensure benchmark quality through human verification. Our extensive experiments reveal that even the best-performing models achieve an accuracy of only 65.3%, underscoring the substantial need for more powerful omni-modal models with enhanced audio-visual fusion capabilities.

## ACKNOWLEDGMENTS

This work is supported by the National Natural Science Foundation of China (No. 62276268) and Baichuan Inc. We also gratefully acknowledge the insightful comments and suggestions provided by the anonymous reviewers.

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

# A   MORE DETAILS ON JOINTAVBENCH

## A.1   TASK DEFINITION

We construct a taxonomy of 15 tasks requiring audio-visual joint reasoning ability based on the benchmark's requirements. The detailed task descriptions with examples are presented in Table 5. Since single-scene tasks and multi-scene tasks require focus on movie details (e.g., visual cues, temporal relationships), question templates are designed to ensure the generated questions both capture critical information and comply with task specifications.

Table 5: Problem Categories and Definitions of JointAVBench's Taxonomy

| Task Name | Code | Category Description | Question Example |
|---|---|---|---|
| Speech-based Timepoint Localization | STL | Locate the temporal position of an object in the dialogue | Which objects are mentioned only in the dialogue but not clearly shown in the video, and when does the first object appear in the dialogue? |
| Speaker Spatial Localization | SPL | Locate the spatial position of a character in the video | Where's the character that says Ï'm gonna give you a compliment now"with a contemptuous tone located in the video? |
| Sounding Object Grounding | SOOG | Locate the spatial position of a sound-emitting object in the movie | What is the spatial position of the object that produced the loud bang in the scene? |
| Sound Event Recognition | SOER | Infer what action occurred that caused the sound | What makes the high-pitched, bright sound? |
| Speaker Emotion Recognition | SPER | Identify the emotions of characters | What's the emotion of the speaker that wears a brown jacket over a blue shirt? |
| Musical Tone Inference | MPTI | Determine the overall tone of a scene | What is the overall atmosphere of the scene? |
| Vision-Speech Sequence Recognition | VSSR | Determine the chronological order of element appearances | In what order were the following items mentioned in the video? (a) 'Did you see this guy smoking pot?' (b) The police officer holding an object. (c) 'Excuse me, sir.' |
| Cross-scene Association | CSA | Identify associations between elements across different scenes | Which dialogue in the remaining parts is most relevant to what the man does in 18.56s-35.16s? |
| Multi-plot Ordering | MPO | Order different audio-visual details across different movie segments | In what order were the following items mentioned in the video? (a) The girl is seen adjusting an oxygen mask on a child (b) The woman is seen trimming flower stems. (c) The man says, "This isn't funny anymore." |
| Plot Temporal Grounding | PTG | Identify the approximate temporal position of plot segments | When did the woman in the green tulle outfit reveal her success in the music industry? |
| Audio Function Analysis | AFA | Analyze the purpose of sound effects and background music in movie segments | How does the video depict the movement of the man in the beige suit? |
| Plot Development Prediction | PDP | Predict future plot developments based on existing plot elements | Which of the following options is most likely to occur after this video ends? |
| Audio-Visual Detail Memory | AVDM | Test the model's long-term memory capability | What was the man doing when the police officer asked if they were smoking? |
| Musical Emotion Shift Inference | MESI | Identify emotional changes and trends throughout the movie | How does the emotional tone evolve from the beginning to the middle of the movie? |
| Character Relationship Inference | CRI | Infer complex relationships between characters based on the overall plot | What is the relationship between the man in the brown jacket and the police officer? |

## A.2   MORE STATISTICS

To further analyze the descriptive characteristics of audiovisual events in our dataset, we performed lexical frequency analysis on both captions and QA pairs, visualized through the word cloud in Figure 8c. The results highlight frequent visual descriptors (e.g., "position", "object", "location") and auditory terms (e.g., "sound", "speaker", "speech"). Furthermore, we examined the distribution of question lengths (in words), as illustrated in Figure 8d, which confirms the conciseness of our formulated questions with minimal redundancy. Figure 8a presents the quantitative distribution of QA pairs across different tasks, revealing that single-scene tasks exhibit particularly rich audiovisual

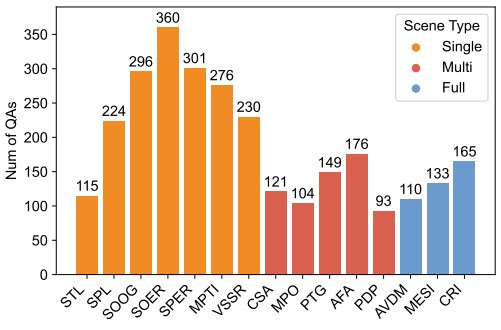

(a) The number of MCQs across tasks.

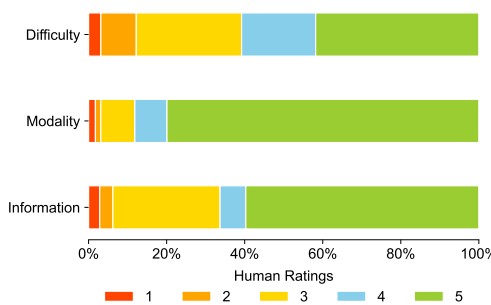

(b) The distribution of human rating scores.

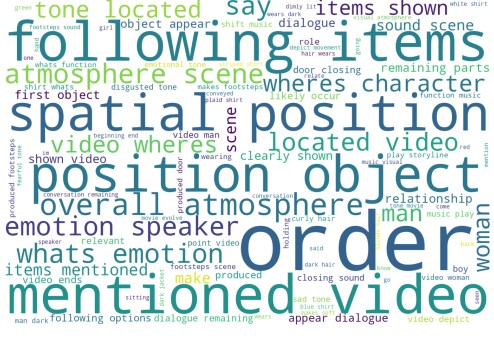

(c) The word cloud presentation of our benchmark.

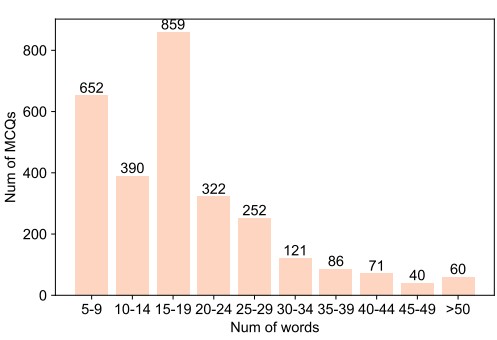

(d) The distribution of word count.

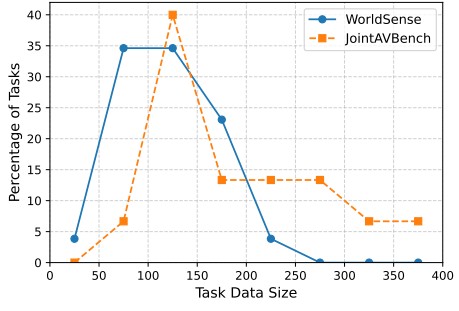

(e) A comparison of task distribution

Figure 8: More details of JointAVBench. The number 1-5 in Figure (b) are human ratings, 1 represents the lowest score and 5 represents the highest score.

correlations and simplicity in question designs. We also present the comparison of our task sample size with that of WorldSense (Hong et al., 2025) in Figure 8e. This comparison indicates that our benchmark contains more data samples per task compared with WorldSense, which also proves our benchmark's credibility.

## A.3 MANUAL CHECK

To ensure the quality of JointAVBench, we engage a group of annotators to evaluate and correct the questions according to the criteria specified in Section 3.3. The annotation process consists of two main steps: First, annotators rate the generated MCQs; then, for incorrect MCQs where the correct answer appears in the distractors, they replace the designated correct answer with the appropriate distractor. Through this process, we maintain high-quality MCQs while discarding only those that fail

to meet our quality standards. After human verification, we discard 17.4% of the data for incorrect answers. Other data are filtered based on the other 3 judging criteria, resulting in 2,853 MCQs.

### A.4 POST-AUDIT ANSWER REFINEMENT

After manual verification, we conduct an additional answer-label audit on the retained benchmark. This step is performed after the human verification stage: human verification determines which samples are retained, while the post-audit step only refines potential answer-label inconsistencies within the retained set. For each MCQ, we ask a strong external LLM auditor to independently select one answer from the given question and options, without using outputs from any evaluated model. We then combine the auditor's answer with the original human annotation through majority voting. When no majority agreement is reached, we retain the original human-verified label. This step is used only to refine likely answer-label inconsistencies; it does not remove samples or change the benchmark size. All reported experimental results are recomputed using the audited labels.

Figure 8b presents the human evaluation scores of the automatically generated MCQs. The results demonstrate that our pipeline generates high-quality questions, with the majority of MCQs receiving the highest rating across all evaluation criteria. Specifically, we find that our QAs are relatively difficult, with the difficulty criteria having high ratings.

### A.5 ANNOTATION INFORMATION

To ensure that our manual data check process is rigorous, we have selected professional annotators. Our annotation team is provided by a professional crowdsourcing labeling company. Each of our annotators has a bachelor's degree from a highly ranked university, is proficient in English, and has passed a qualification test on sample tasks before participating.

### A.6 EXPERIMENT DETAILS

All reported results are computed on the audited benchmark described in Appendix A.4. To ensure comparability between MLLMs, we selected similar model parameters and video frames. For open-source models, we selected 7B as the parameter size (we selected InternVL2.5-8B (Chen et al., 2024c) since it does not provide a 7B model). We also selected 32 frames as the maximum number of frames per video to ensure that the frame number did not affect performance. For closed-source models, we adhered to the official model settings. Notably, for GPT-4o (Hurst et al., 2024), we only input video frames alongside the question text, and therefore, the model is unable to access timestamps. During the experiment, we randomly shuffled the options to ensure that the distribution of correct answer prefixes was uniform and free from biases. For Gemini2.5-Flash, we use the setting: temperature=1.0, max_temperature=2.0, top_p=0.95, top_k=64, thinking=True. For Gemini2.5-Pro, we use the setting: temperature=1.0, max_temperature=2.0, top_p=0.95, top_k=64, thinking=True.

All experiments were conducted on NVIDIA H-100 GPUs and can be reproduced using a single H-100 (80G) GPU. During the automatic generation process, we used the official API (model name: 'qwen2.5-72b-instruct') from Qwen to ensure long-context capability and generation stability.

### A.7 CASES

Figure 9 presents detailed examples from our benchmark. The questions and options in Joint-AVBench are designed to incorporate both audio and video modalities and to evaluate audio-visual joint reasoning capabilities. We present the task names, a few frames of the video, questions, options, and correct answers in the cases.

## B DETAILS ON GENERATION PIPELINE

### B.1 VIDEO CAPTION GENERATION

We generate visual captions using Qwen2.5-VL (Bai et al., 2025) at a rate of 1 frame per second (fps) for each identified scene, regardless of its quality level (low or high). To ensure the captions capture

both static and dynamic scene information, we carefully design a video captioning prompt as shown in Figure 10.

## B.2 AUDIO CAPTION GENERATION

To ensure the audio captions contain rich information about diverse audio signal types, we generate separate captions for each type. We then utilize an LLM with carefully designed Chain-of-Thought (CoT) prompts to reduce caption hallucination.

**Vocal Traits, Sound Event, and Music Description.** We find that directly using general-purpose audio-language models (ALMs) to generate overall audio captions often overlooks important details and may produce inaccurate results. Therefore, we employ Qwen2.5-Omni (Xu et al., 2025a), an open-source multimodal model, to separately generate descriptions of vocal traits, sound events, and music components. Notably, current ALMs cannot reliably distinguish between sound events and music. We address this limitation by generating both sound event and music descriptions initially, then separating them during post-processing. The detailed prompt templates are provided in Figure 10.

**Subtitle Transcription.** For dialogue transcription, our primary objectives are accurate speech recognition and precise timestamp generation. Since general ALMs underperform in timestamp estimation, we utilize Whisper-v3 (Radford et al., 2023), an advanced and widely utilized automatic speech recognition (ASR) system, to ensure transcription quality.

**Audio Caption Refinement.** The initial audio descriptions contain hallucinations (*i.e.*, factually incorrect or repetitive outputs). We employ Qwen-2.5 (Yang et al., 2024) to perform three refinement steps: (1) distinguishing between background music and sound events, (2) aligning vocal characteristics with dialogue transcripts, and (3) removing redundant content. The detailed prompt engineering for this process is illustrated in Figure 11.

## B.3 QA PAIR GENERATION

We utilize Qwen2.5 (Yang et al., 2024) to generate QA pairs following predefined question templates. For single-scene and multi-scene tasks, we only provide descriptions for high-quality scenes. For full-scene tasks, we include all scene descriptions regardless of quality to ensure no details are omitted. After generating multi-scene QA pairs, we verify the interval between questions to ensure they require information from multiple scenes, using the prompt shown in Figure 16 to identify their information intervals. Additionally, we require the model to generate a brief justification for each answer while generating QA pairs.

## B.4 QUALITY CONTROL

In the quality control stage, we employ extensive CoT techniques to ensure that Qwen2.5 (Yang et al., 2024) achieves optimal performance in identifying potential hallucinations.

During the general check, we utilize only the QA pair and its explanation to filter out unqualified QA pairs. This stage includes four checks: modality, format, content, and speculation checks. The details of each check are as follows: (i) modality check assesses whether the modality clues used in the QA pair are derived from dual modalities; (ii) format check evaluates whether the answer corresponds to the question in format (*e.g.*, the answer explains two items, but the question asks about only one item); (iii) content check verifies whether the answer can be logically inferred from the question based on the explanation; (iiii) speculation check examines whether the answer relies excessively on speculation rather than concrete evidence. The prompts used in this stage are shown in Figure 13.

For the specific check, we design task-specific prompts based on the definition of each task. The prompts for each specific check are presented in Figure 14 and Figure 15. Note that for better adaptation to different tasks, the prompt for the sequence check varies slightly between VSSR and MPO, and the ambiguity check varies slightly between SPL and SOER.

## B.5 DISTRACTOR GENERATION

Since generating distractors requires additional information from the video, we generated them after filtering the QA pairs. In this process, we designed a generation prompt incorporating various error

Table 6: AV Correlation Scores for Different Datasets

| Dataset | Automatic Score | Human Score |
|---|---|---|
| Music-AVQA | 56.7% | 54.5% |
| OmniBench | 80.4% | 94.0% |
| AV-Odyssey | 99% | 99.5% |
| LongVALE | 76.2% | 75.0% |
| WorldSense | 62.9% | 60.5% |
| JointAVBench (ours) | 93.5% | 94.5% |

Table 7: Examples of failure cases.

| Task | Question | Groundtruth Answer | Model Answer |
|---|---|---|---|
| SPL | Where's the character that says 'Oh, my God' with a vibrant shocked tone located in the video? A. In the left B. In the center C. Standing behind the loveseat D. On the right | A | D. On the right. |
| SPER | What's the emotion of the speaker that wears a vibrant green tulle outfit with hair rollers and dramatic makeup? A. Surprised B. Amused C. Confused D. Excited | D | A. Surprised. |
| MPO | In what order were the following items mentioned in the video? (a) 'When do you need to have the van back'. (b) The man talked about cars with a joyful tone (c) The man is driving a car along a road lined with bare trees A. (c) (b) (a) B. (a) (b) (c) C. (b) (c) (a) D. (c) (a) (b) | D | The correct answer is A. |
| PTG | When did the woman in the green tulle outfit reveal her success in the music industry? A. 51.05s-80.79s B. 126.38s-174.51s C. 86.46s-126.38s D. 45.84s-51.05s | A | The correct answer is B. |

types to ensure option diversity and complexity, as illustrated in Figure 16. This includes common error categories such as incorrect details and temporal/spatial misplacement.

## C  MORE EXPERIMENTS

### C.1  EVALUATION OF AV CORRELATION FOR PREVIOUS WORKS

We evaluate the AV correlation ratio (the proportion of questions that truly require auditory and visual information to answer) for previous works in Table 1. However, since some benchmarks do not evaluate the ratio themselves, we utilize Qwen-2.5 (Yang et al., 2024) to judge the correlation. We utilize our modality check prompt (shown in Figure 13) and sample 1,000 QA pairs from each dataset to judge. We also recruit human volunteers to verify the AV correlation ratio by asking them to assess whether the randomly sampled 100 data items from each benchmark require both auditory and visual information to answer. The overall comparison is described in Table 6, indicating that our automatically calculated score has high correspondence to human scores. This score indicates that our benchmark can assess joint audio-visual reasoning ability.

### C.2  ERROR STUDY

We'd like to provide some failure cases and deep analysis in this section. The failure cases are selected from the worst-performing tasks.

Based on the failure cases, we find that:

- **Models fail to understand vocal traits.** In the first example, the model fails to find the spatial information based on speech and vocal traits. In the second example, the model can't find vocal traits based on visual information. These two examples indicate that the ability to align visual information with vocal traits remains low in current models and needs to be improved.

- **Models fail to understand temporal information.** In the third example, the question tests the model's ability to understand the storyline and arrange the detailed information. And

the fourth example tests the model's temporal grounding ability from a long video. These two examples showcase that future works should focus on increasing the model's ability to understand temporal relationships in audio-visual scenarios.

## D    LIMITATIONS

We acknowledge that the JointAVBench has several limitations in its generation and experimental evaluation. First, the dataset is exclusively derived from SF20K, which may introduce biases in data distribution. Second, our designed taxonomy, while comprehensive, may not encompass all dimensions of audio-visual joint reasoning capabilities. Nevertheless, we have rigorously ensured that the included tasks cover critical aspects and effectively assess the target abilities. Third, due to computational constraints, our experiments were limited to selected representative MLLMs rather than an exhaustive evaluation. Moreover, we admit that our dataset is constructed through an automatic pipeline, which can introduce quality risks from imperfect captions, LLM-based filtering, distractor generation, and ambiguous long-video evidence. Although we conduct manual verification and an additional answer-label audit, these steps may still miss subtle factual errors, weak audio-visual dependency, or cases where multiple options are partially plausible. Finally, we acknowledge the possibility that proprietary models may have seen overlapping content; however, no public training data disclosure allows verification. We intend to address these limitations in future work through dataset expansion and more extensive benchmarking.

## E    BROADER IMPACTS

We constructed JointAVBench to facilitate research and development in omni-LLMs and video understanding. We anticipate that this dataset may yield both positive and negative societal impacts. JointAVBench offers several potential benefits, including: (1) enabling development of human-like agent systems, (2) advancing video understanding tools, and (3) creating assistive software for people with disabilities. However, the dataset also presents certain risks, such as privacy concerns and copyright issues. We believe a thorough discussion of these benefits and challenges will lead to a more comprehensive understanding of the dataset's societal implications.

## F    DECLARATION OF LLM USAGE

During our research, we use LLMs as a major dataset construction tool, including dataset generation, quality control, and experiments. During our paper writing, we use LLMs to polish our paper and correct the defects in our paper.

## G    SAFEGUARDS

To ensure our benchmark excludes unsafe content, we adopted two key measures. First, we used the publicly released SF20K dataset (Ghermi et al., 2024) as the foundation, which provides pre-filtered safe content. Second, we employed the official Qwen API during benchmark generation, whose built-in safety mechanisms automatically screen both input prompts and output responses for potentially unsafe video recommendations.

## H    LICENSE

The JointAVBench dataset is released under the CC BY-NC-SA 4.0 license. Subsequent research using this dataset must comply with the license terms.

### Speech-based Timepoint Localization

**Question:**
Which objects are mentioned only in the dialogue but not clearly shown in the video, and when does the first object appear in the dialogue?
**A**. The frisbee, mentioned at around 329s
**B**. The pig in a blanket, mentioned at around 329s
**C**. The green box, mentioned at around 502s
**D**. The dead porcupine, mentioned at around 445s
**Correct Answer**: A

### Sounding Object Grounding

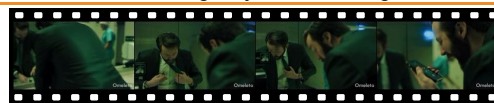

**Question:**
Where's the sounding object located in the video?
**A**. Near the sink.
**B**. By the exit door.
**C**. In the hallway.
**D**. Near the stalls.
**Correct Answer**: D

### Sound Event Recognition

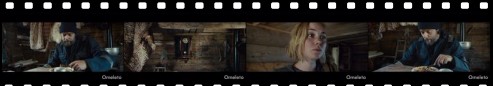

**Question:**
What makes the footsteps sound?
**A.** The man getting up from the table
**B.** The door closing as the woman enters the cabin
**C.** The woman approaching the man
**D.** The woman pacing back and forth in the cabin
**Correct Answer**: C

### Musical Tone Inference

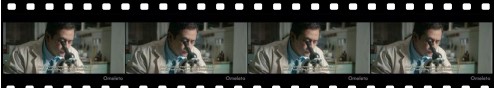

**Question:**
What is the overall atmosphere of the scene?
**A.** Busy and chaotic
**B.** Calm and relaxed
**C.** Somber and focused
**D.** Bright and cheerful
**Correct Answer**: C

### Plot Temporal Grounding

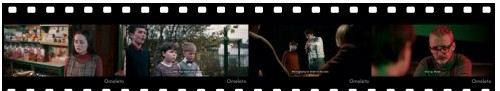

**Question**:
At what point in the video does the boy in the orange sweater explain to the man in the dark turtleneck that the money he brought is his own, not his father's?
**A.** 668.04s-710.88s
**B.** 778.08s-816.42s
**C.** 759.88s-772.00s
**D.** 744.29s-751.62s
**Correct Answer**: B

### Cross-scene Association

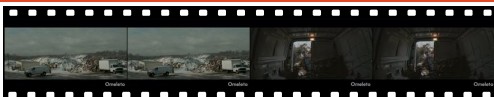

**Question**:
Which dialogue in the remaining parts is most relevant to what the man does in 18.56s-35.16s?
**A.** "Excuse me."
**B.** "I'll take care of them."
**C.** "This is true."
**D.** "Stop."
**Correct Answer**: B

### Charater Relationship Inference

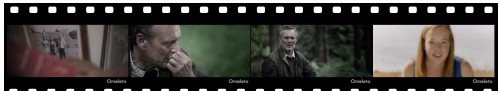

**Question**:
What is the relationship between the man with the blood-stained hands and the woman in the red jacket?
**A.** The man and the woman are father and daughter.
**B.** The man and the woman are brother and sister.
**C.** The man and the woman are uncle and niece.
**D.** The man and the woman are husband and wife.
**Correct Answer**: A

### Musical Emotion Shift Inference

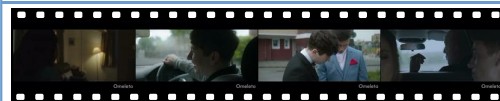

**Question**:
What is the emotional shift depicted in the scenes leading up to and including the car ride after the "debs"?
**A.** From celebratory and joyful to tense and anxious.
**B.** From calm and serene to lively and energetic.
**C.** From celebratory and joyful to reflective and tender.
**D.** From reflective and tender to celebratory and joyful.
**Correct Answer**: C

Figure 9: Additional cases of JointAVBench. The first and second row represents single-scene tasks, the third row represents multi-scene tasks, and the last fourth row represents full-scene tasks.

**Prompt for video captioning**

Please generate a clear, concise, and detailed caption for the provided movie video clip. Ensure the description is entirely based on the visual content of the video, without any speculation, assumptions, or uncertain information. Focus on capturing the most essential visual elements while avoiding unnecessary repetition or overly verbose language. Describe the scene as follows:
1. Scene Setting: Briefly describe the environment, location, time of day, lighting, and any notable objects or elements in the background. Only include details that are clearly visible in the video.
2. Characters and Actions: Highlight the appearance, clothing, and key actions of any characters present. Focus on their most significant movements, gestures, and interactions. Do not infer emotions, intentions, or backstory unless explicitly shown through visual cues.
3. Scene Dynamics: Describe any important changes in the scene. Include the sequence of events and the pacing of the scene to convey how it unfolds over time. Only describe what is visually evident.
4. Emotional Tone: Convey the mood or atmosphere through the most impactful visual cues, such as facial expressions, body language, or environmental details. Only describe emotions that are clearly expressed through visible actions or expressions.
5. Key Events: Highlight any significant events or actions that occur within the scene, focusing on their narrative importance or impact on the characters. Only include events that are explicitly shown in the video.
Important Notes:
- Strictly factual: Ensure the description is entirely based on the visual content of the video. Do not include any information that is not explicitly shown or clearly visible.
- Avoid speculation: Do not infer or assume anything about characters' thoughts, feelings, or motivations unless they are directly expressed through visible actions or expressions.
- Acknowledge uncertainty: If certain details are unclear or ambiguous, do not include them in the description. Focus only on what is clearly visible.
- Exclude text: If there are subtitles or text displayed at the bottom of the video, do not include them in the caption. Focus only on the visual elements of the scene.
- Be concise: Avoid unnecessary details or repetition. Focus on the most critical visual elements that define the scene.
Use vivid but economical language to paint a clear and engaging picture of the scene for someone who cannot see the video.

**Prompt for sound event & music captioning**

Describe the audio by focusing ONLY on the following detectable elements:
1. Sound Events (if clearly present):
- Non-speech sounds: (e.g., crumpling, footsteps, door closing, glass breaking).
- Non-verbal vocalizations: (e.g., laughter, sighing, coughing, crying, humming, screaming).
- Characteristics (only if unambiguous):
- Pitch (high/low), timbre (bright/muffled), rhythm (steady/erratic), volume (loud/soft).
- Rules:
- Do NOT guess sound sources (e.g., no "paper crumpling" → just "a crumpling sound").
- If uncertain, omit the detail entirely.
2. Background Music (if clearly present):
- Instruments (e.g., piano, strings, electric guitar).
- Mood(e.g., cheerful, tense, melancholic).
- Avoid technical terms (BPM, key, scales).
- Rules:
- Do NOT describe lyrics or vocal melodies.
- If the music is ambiguous (e.g., genre unclear), only state observable features.
Critical Constraints:
- Do NOT describe speech, spoken words, conversation content, or verbal interactions.
- Never use speculative language (e.g., "likely," "probably," "seems like").
- If a sound cannot be confidently identified, skip it.
- No timestamps, durations, or speaker demographics (e.g., "a child laughing" → just "laughter").

**Prompt for vocal traits captioning**

Please analyze the provided speech audio and generate a strictly factual description following these requirements:
1. Output format for each utterance:
Speech Content: [Exact dialogue content]
Emotion: [Observed emotional tone] (eg. happy, sad, angry, fearful, surprised, disgusted, excited)
Speaker traits: [Directly discernible characteristics like age/gender if evident from voice]
2. Rules:
(1) Only describe emotions clearly conveyed through vocal tone
(2) Note speaker characteristics ONLY when immediately apparent from voice (e.g. "child-like voice")
(3) Never add interpretations beyond what the audio contains
(4) Process each utterance separately
(5) Non-speech audio (music or sound only): output "[Non-speech audio: skip analysis]" and stop

Figure 10: Prompts for generating omni-modal caption.

**Prompt for cleaning and separating sound event caption and music caption**

You are an expert in audio data cleaning. Your task is to clean and organize audio captions by separating them into two distinct categories: [music] and [sound event].

Please analyze the provided audio caption and separate it into two parts:

`[music]`: Include descriptions related to music, such as background music, musical instruments, and any emotional or atmospheric content (e.g., tense, sad). If no music is described, output "None".

`[sound event]`: Include descriptions of Non-musical sounds (e.g., actions, engine noises, wind) and Non-verbal vocal sounds (e.g., laughter, sighing, coughing) (EXCEPT for any mention of "Crumpling sound" - these should be completely omitted). If no sound events are described (after filtering out Crumpling sounds), output "None".

Do not include any content related to speech, dialogue, or verbal communication in either section. Remember not to generate any explanations or notations after the separated part output.

Output Format:

[music] ¡description of music and emotional characteristics if any¿ or None

[sound event] ¡description of sound events¿ or None

Example: ¡example¿

---

**Prompt for cleaning vocal traits caption**

You are an expert in speech and subtitle data analysis. You have access to the following:

Vocal traits: The text that describes the emotion and other characteristics of speech sentence by sentence, each utterance contains speech content, emotion, and speaker traits.

Subtitle: The transcription of speech. Each line in the subtitle represents a single phrase.

Your task is to evaluate the provided vocal traits by comparing them with the corresponding subtitle. The goal is to determine whether the vocal traits contain usable information based on their alignment with the subtitle. Your analysis should be rigorous and follow a structured approach.

Please follow the guidelines below:

1. Comparison with Subtitle:

- Directly compare the Speech Content text with the provided subtitle.

- Discard any speech content marked with neutral emotion (e.g., "neutral tone", "neutral mood") immediately, regardless of subtitle alignment.

- For non-neutral speech content:

- If the speech content contains phrases that overlap or align with the subtitle (e.g., shared keywords or contextual similarity), proceed to the next stage.

- If no alignment is found, the vocal traits are unusable.

2. Output of Emotional Information:

For each utterance:

- Only if the vocal traits align with the subtitle and are non-neutral:

- Replace any speech content in the vocal traits (i.e., quoted or referenced dialogue) with the exact matching phrase from the subtitle.

- Preserve all other emotional/tonal descriptors (e.g., "sad mood," "English accent").

- Format the output as a coherent description combining the subtitle content and emotional features.

- If no alignment exists, output '[Unavailable]'.

- Always prefix the final output with '[Output]' or '[Unavailable]'.

Provide the final cleaned utterances in the following format:

[Output] ¡utterances with emotional/tonal information¿ if subtitle roughly matches or [Unavailable] if NO utterance is available

Example: ¡example¿

Input:

Vocal Traits: {vocal_traits}

Subtitle: {subtitle}

Please follow the guidelines to clean the vocal traits text, and provide the final output in the specified format. Remember to output '[Unavailable]' only when no utterance is available.

Figure 11: Prompt for audio caption refinement.

**Prompts for generating single-scene QA pairs**

Your task is to generate a question-answer pair based on the instructions. The question must utilize both visual and audio information (e.g., speech, sound event, or music). Generate a question-answer pair based on this analysis, ensuring the question does not contain any hints or details about the answer. The answer should be precise and concise, and include an explanation justifying the answer. If the material does not provide sufficient information to generate a valid question-answer pair, respond with '[Unavailable]'.
Format the output as follows:
`[Question]`......
`[Answer]`......
`[Explanation]`......
Please follow the instructions below:{instruction}
Example:{examples}
Here are the input material:{input_modality}
Please follow the instructions and refer to the examples provided to assist with your question design.

**Prompts for generating multi-scene QA pairs**

Your task is to generate five different question-answer pairs based on the instructions. Each question must integrate information from both the audio and video modalities, ensuring that neither modality alone can provide the answer. Importantly, the information used to formulate each question should be derived from a few consecutive segments of the material, rather than from a single segment or the entire content.
The generated question-answer pairs should be unique and avoid overlapping in content. The questions should be designed without revealing hints or details about the answers. The answers should be precise and concise, accompanied by a brief explanation that justifies the response based on the combined audio-visual information from the selected segments. Use the video description to enhance your understanding of the material.
Format the output as follows:
¡Output id¿
`[Question]`......
`[Answer]`......
`[Explanation]`......
Please follow the instructions below:
Generate five different question-answer pairs that {instruction}
Example:{examples}
Here is the input material:{segments_info}
Please follow these instructions and refer to the examples provided to guide your question design.

**Prompts for generating full-scene QA pairs**

Your task is to generate five different question-answer pairs based on the instructions. The questions must utilize both visual and audio information, and cannot be answered by information from only one modality. The five generated questions should be different from each other. The questions should also be derived based on the whole movie, not just a few segments. Generate the question-answer pairs based on this analysis, ensuring each question does not contain any hints or details about the answer. The answers should be precise and concise, and include an explanation justifying the answer for each question. You can use the video description to help you better understand the material.
Format the output as follows:
¡Output id¿
`[Question]`......
`[Answer]`......
`[Explanation]`......
Please follow the instructions below:
Generate five different question-answer pairs that {instruction}
Example:{examples}
Here are the input material:{segments_info}
Please follow the instructions and refer to the examples provided to assist with your question design.

Figure 12: Prompts for generating QA pairs

**Prompt for modality check in general check**

You are a multimodal understanding assistant. You have access to the following:
1. Question: A question related to the video clip.
2. Answer: An answer provided to the question.
3. Explanation: An explanation supporting the answer.
Your task is to evaluate whether the question can be answered using only one modality (either video or audio) or if it requires both modalities. Please strictly base your judgment on the information explicitly required to answer the question, as well as the content of the provided answer and explanation. Avoid making assumptions about the content of the modalities beyond what is explicitly stated in the question, answer, or explanation.
Please follow these steps to complete your evaluation:
1. Information Analysis:
Analyze the question to identify the specific visual and auditory details required to answer it. Does the question require visual details (e.g., objects, actions, or settings) or auditory details (e.g., speech, sound effects, or music)?
Extract the visual and the auditory information from the explanation to determine which modalities are used to support the answer.
2. Modality Assessment:
Based on the analysis of the question and explanation, determine if the required information can be obtained entirely from one modality (either video or audio) or if both audio and visual modalities are necessary.
- Determine if the question can be answered using only the extracted video text.
- Determine if the question can be answered using only the extracted audio text.
If the question requires information from both the video text and the audio text to be answered, then it is considered feasible to use both modalities.
3. Conclusion:
Provide your final determination: Output [YES] if the question explicitly requires information from both video and audio modalities to be answered correctly, or if the answer and explanation rely on information from both modalities. Otherwise, output [NO]. Here is the question: {question}
Here is the answer: {answer}
Here is the explanation: {explanation}
Please complete the Information Analysis, Modality Assessment, and Conclusion stages with the special answer token [YES] or [NO].

**Prompt for checking format content and speculation in general check**

You are a quality evaluation assistant. You have access to the following:
1. Question: A question related to a given context.
2. Answer: An answer provided to the question.
3. Explanation: An explanation supporting the answer.
Your task is to evaluate the quality of the question-answer pair by performing two checks: format check and content check. Please strictly base your judgment on the information explicitly provided in the question, answer, and explanation. Avoid making assumptions beyond what is stated. Please follow these steps to complete your evaluation:
1. Format Check:
- Analyze the question to determine how many distinct pieces of information it is asking for.
- Check if the answer addresses all the pieces of information requested in the question.
- If the question asks for only one piece of information and the answer fully addresses it, proceed to the content check.
- If the question asks for multiple pieces of information but the answer only addresses some of them, output '[NO]' and stop.
2. Content Check:
- Analyze the explanation to determine if it is reasonable and logically sound.
- Check if the answer can be derived from the explanation and if the answer is correct based on the context of the question.
- If the explanation is reasonable and the answer is correct and supported by the explanation, output '[YES]'.
- If the explanation is unreasonable or the answer cannot be derived from the explanation, output '[NO]'.
3. Speculation Check:
- Analyze the explanation to determine if the answer relies too heavily on speculation rather than concrete evidence or logical reasoning.
- If the explanation provides clear, evidence-based reasoning or logical steps to derive the answer, proceed to the final output.
- If the explanation relies on assumptions, guesses, or unsupported claims, output [NO] and stop.
Final Output:
- If both the format check and content check pass, output '[YES]'.
- If either check fails, output '[NO]'.
Here is the question: {question} Here is the answer: {answer} Here is the explanation: {explanation} Please complete the Format Check, Content Check, and Final Output stages with the special answer token '[YES]' or '[NO]'.

Figure 13: Prompts for general checks

---

**Prompt for sequence check**

You are a quality control assistant for video-based question-answering pairs. Your task is to validate whether a given QA pair about element ordering in videos is correct and properly sourced. Follow this 4-stage process:

Stage 1: Element Extraction
- Extract all elements to be sorted from the question (format: (a)[element1], (b)[element2], etc.)
- Verify exactly 3 elements exist. If not, immediately output "[NO]"

Stage 2: Occurrence Localization
For each extracted element:
1. Search through all video segments to find the first occurrence
2. Record for each element:
- Segment ID of first appearance
- Modality type (video caption/subtitle/speech emotion)
3. If any element cannot be found → Output "[NO] (unverifiable element: [element_name])"

Stage 3: Element Validation
Verify the located elements meet these criteria:
1. Unique Segment Check:
- All elements must appear in different segments
- If any segment ID is shared → Output "[NO] (co-occurring elements: [element1] & [element2] in segment X)"
2. Modality Diversity Check:
- Elements must come from ≥ 2 different modalities
- If all same modality → Output "[NO] (single modality: [modality_type])"

Stage 4: Order Verification
1. Sort elements by their first appearance segment ID (ascending)
2. Compare against the provided answer:
- If orders match → Output "[YES]"
- If orders differ → Output "[Corrected]" with proper order and explanation

Output Format:
`[Validating]` ¡4-stage analysis¿
`[Output]`: [YES/NO/Corrected]
`[Corrected:  (a)  (b)  (c)]` (if applicable)
`[Explanation]` (if Corrected):
- (a) [element1]: first appears in segment [X] ([modality])
- (b) [element2]: first appears in segment [Y] ([modality])
- (c) [element3]: first appears in segment [Z] ([modality])
Provided Information:
Video Text: {segments_info}
Question: {question}
Answer: {answer}
Explanation: {explanation}
Please perform the stages above carefully in [Validating] and provide the final output in the specified format.

---

**Prompt for ambiguity check**

You are a QA evaluation assistant tasked with filtering incorrect or low-quality question-answer pairs based on video and audio context. Follow this structured evaluation:

Phase 1: Specificity Check
- Check if the 'answer' is overly generic (e.g., fails to distinguish between objects/agents).
- Example: If the 'question' asks "Where is the sound source located?" and the 'answer' is "in a wooden house" (while the entire video occurs in a wooden house), mark as "[NO]" (lacks specificity).
- Output: Proceed only if "[PASS]"; else, output "[NO]".

Phase 2: Sound Source Ambiguity (Video Context)
- Using the 'Video Caption', verify if other objects in the scene could plausibly produce the sound mentioned in the 'question'.
- Example: If the 'question' asks "What caused the splashing sound?" and the 'Video Caption' only describes a "person by a pool," but no other water-related objects exist, mark as "[NO]".
- Output: Proceed only if "[PASS]"; else, output "[NO]".

Phase 3: Cross-Modality Dependency
- Determine if the 'answer' can be derived solely from 'Video Caption' + 'question' + commonsense (ignoring 'Sound Event').
- Example: If the 'question' asks "Where is the splashing sound coming from?" and the 'Video Caption' mentions "a beach with waves," commonsense suggests "ocean" → mark as "[NO]" (audio not needed).
- If 'Sound Event' is required (e.g., to distinguish between similar objects), mark as "[PASS]".
- Output: If "[PASS]" in all phases, output "[YES]"; else, "[NO]".

Final Output:
- Only "[YES]" or "[NO]" based on the above checks.
Provided Information:
{segments_info}
Here is the question-answer pair to be analyzed:
- Question: {question}
- Answer: {answer}
- Explanation: {explanation}
Perform the analysis step-by-step and output either "[YES]" or "[NO]" based on your evaluation.

Figure 14: Prompts for sequence check and ambiguity check.

**Prompt for sound event check**

You are a QA evaluation assistant tasked with filtering incorrect question-answer pairs based on video and sound event information. Follow this phased approach:
Phase 1: Off-Screen Sound Check
- If the Answer describes an off-screen sound event (e.g., "An off-screen object"), output [NO].
Phase 2: Sound Event Presence Validation
- Extract the sound event mentioned in the Question, Answer, or Explanation.
- Check if this sound event exists in the provided Audio Text. If not, output [NO].
Phase 3: Contextual Consistency with Video
- Using only the Video Caption (ignore Explanation), verify if the sound event logically fits the scene.
- Example: If the sound event is "glass breaking" but the Video Caption lacks glass-related objects/actions, output [NO].
- Do not speculate; reject if the video lacks supporting evidence.
Phase 4: Final Judgment
- If all phases pass, output [YES]. Otherwise, output [NO].
Here is the provided information:{segments_info}
Question: {question}
Answer: {answer}
Explanation: {explanation}
Perform the four-phase analysis and output either '[YES]' or '[NO]'.

**Prompt for music check**

You are a QA pair evaluation assistant. Your task is to determine whether a given question-answer pair is valid based on the provided video description and music content. Follow these steps:
1. Phase 1: Music Information Validation
- Extract the music-related information used in the 'Answer' and 'Explanation' of the QA pair.
- Check if this music information appears in the provided 'Music Content'.
- If the music information is not found in the 'Music Content', output '[NO]' (invalid QA pair).
2. Phase 2: Visual Information Cross-Check
- If the music information is valid (from Phase 1), analyze whether the emotion/atmosphere described in the music can also be inferred from the 'Video Caption' (e.g., character expressions, scene mood, or events).
- Example: If the music mentions a "sad atmosphere" and the video shows "a character crying," the music info can be inferred visually → '[NO]'; If the music describes a "warm atmosphere" and the video caption mentions "bright lighting," the music info can be inferred visually → '[NO]'; If the music's emotional/atmospheric cues can be derived from the video alone, output '[NO]'.
3. Phase 3: Final Judgment
- If the QA pair passes both Phase 1 and Phase 2 (i.e., music info is valid and cannot be inferred visually), output '[YES]'.
- Otherwise, output '[NO]'.
Here is the provided information:{segments_info}
- Question: {question}
- Answer: {answer}
- Explanation: {explanation}
Perform the three-phase analysis and output either '[YES]' or '[NO]'.

**Prompt for vocal traits check**

You are an assistant tasked with validating question-answer (QA) pairs generated from video content, specifically focusing on the use of speech emotion data (Speech Content, Emotion, Speaker Traits). Your goal is to filter out incorrect or weakly supported QA pairs by following this phased approach:
Phase 1: Extract Utilized Speech Emotion Information
- From the Question and Explanation (if provided), identify:
- Speech Content: Exact phrases/words from the audio used to answer the question.
- Emotion: The emotion label (e.g., "angry," "joyful") tied to the speech content.
- Speaker Traits: Any speaker characteristics (e.g., "deep voice," "child") referenced.
- Output: List only the explicitly used components. If none are used, stop and output '[NO]'.
Phase 2: Verify Grounding in Provided Speech Emotion Text
- Check if the extracted Speech Content, Emotion, and Speaker Traits from Phase 1 appear verbatim or unambiguously in the provided speech emotion text.
- Example: If the QA pair uses "confused" emotion for "What?", but the speech emotion text lacks this pairing, it fails.
- Output: If any extracted component is missing, output '[NO]'.
Phase 3: Assess Text-Based Emotion Inferrability
- For the Speech Content and Emotion pair used in the QA pair, determine if the emotion could be directly inferred from the text alone (e.g., "I'm furious" → "angry").
- Disqualify: Obvious cases (e.g., sarcasm-free explicit statements, clear interrogatives).
- Output: If inferrable, output '[NO]'.
Phase 4: Check Video-Based Emotion Redundancy
- Determine if the Emotion used in the QA pair could also be clearly deduced from the video caption (e.g., "she frowns" → "sad").
- Note: Assume video captions describe visible emotions unless stated otherwise.
- Output: If deducible, output '[NO]'.
Final Decision
- Only output '[YES]' if all phases are passed (i.e., the QA pair uses non-inferrable, video-independent speech emotion data with explicit grounding).
- For any phase failure, output '[NO]'.
Provided Information:{segments_info}
- Question: {question}
- Answer: {answer}
- Explanation: {explanation}
Perform the analysis step-by-step and output either "[YES]" or "[NO]" based on your evaluation.

Figure 15: Prompts for audio check.

---

**Prompt for interval identification**

You are an expert in finding all the continuous segments needed to answer a question-answer pair. Your task is to identify the minimal continuous sequence of movie segments (neither the first nor last segment) that contains all the necessary information to answer the given question.
You will be provided with:
1. A question-answer pair
2. An explanation of how the answer is derived
3. Complete information about all movie segments (timestamps, video descriptions, audio descriptions, and subtitles)
Instructions:
1. Carefully analyze the question and answer explanation to understand what information is required
2. Examine all movie segments sequentially to locate where the relevant information begins
3. Determine where the last necessary piece of information appears
4. Select the earliest segment where required information starts (start segment) and the latest segment where required information ends (end segment)
Ensure:
1. The selected segments form a continuous sequence
2. The sequence is not from the very first to the very last segment
3. All information needed to answer the question is contained within this sequence
4. The sequence is as compact as possible
Output Format:
Provide your response in this exact format:
`[Start]: ¡segment number¿ [End]: ¡segment number¿ [Rationale]: ¡brief explanation of why these segments were chosen¿`
Important Notes:
- If the answer requires information that only appears in disjoint segments, select the smallest continuous sequence that contains all relevant segments
- The start and end segments must be different (cannot be the same segment)
- Never choose segment 0 and the final segment at the same time
Input:
Here is the question: {question}
Here is the answer: {answer}
Here is the explanation: {explanation}
Here are the segments information:{segments_info}
Please follow the guidelines and use the input material to identify the start segment and end segment for the question-answer pair. Make sure that the generated output follow output format.

---

**Prompt for generating distractors**

You are an expert in generating multi-choice question. Your task is to generate distractors based on the guidelines.
Given the following background information, question, correct answer, and answer rationale, generate three incorrect answer options (distractors) that closely mimic the correct answer in terms of length, format, and style. The distractors should appear reasonable to someone who doesn't fully grasp the concept but must contain subtle errors (factual, logical, or contextual).
Please follow these guidelines to generate distractors:
1. **Selective Modification**: Alter specific elements such as character actions, dialogue, objects, or settings to create plausible yet incorrect options.
2. **Maintain Plausibility**: Ensure each distractor could feasibly occur within the context of the video, making them appear credible based on the visual and audio cues.
3. **Incorporate Diverse Misdirections**:
- **Action Confusion**: Modify or swap character actions or events in ways that fit the context but are incorrect.
- **Dialogue Adjustments**: Propose believable alterations to dialogue or audio cues that didn't actually occur.
- **Object or Setting Misdirection**: Suggest plausible but incorrect details about objects, settings, or visual elements.
- **Speech Emotion Alteration**: Modify the described emotional tone of speech content while keeping the words themselves accurate.
- **Sound Event Manipulation**: Change specific sound effects or environmental audio cues to similar but incorrect versions that could plausibly exist in the context.
- **Musical Atmosphere Shift**: Adjust the described mood or emotional impact of background music to a different but related atmosphere.
4. **Incorporate Partial Truths**: Use true audio-visual details or partial truths within the distractors to add complexity, ensuring these elements do not directly answer the question but make the distractors more compelling.
5. **Avoid Obvious Falsities**: Shift the context or details significantly without creating options that are blatantly wrong or unrelated to the video.
6. **Ensure Distinct Incorrectness**: Craft distractors that will be clearly identifiable as incorrect by someone who has closely watched and listened to the video, challenging their attention to detail.
Requirements for Distractors:
1. Plausibility: Each distractor should seem correct at first glance, matching the tone and structure of the correct answer.
2. Variety: Errors should vary (e.g., minor inaccuracies, flipped terms, oversimplifications, or common misconceptions).
3. Consistency: Maintain the same verb tense, technicality, and formatting as the correct answer. Format the output as follows:
`[Distractor 1] ¡Incorrect but plausible option¿`
`[Distractor 2] ¡Incorrect but plausible option¿`
`[Distractor 3] ¡Incorrect but plausible option¿`
Provided Information:
Background infromation:{segments_info}
Question: {question}
Correct Answer: {answer}
Answer Rationale: {explanation}
Now generate three high-quality distractors for the given question and correct answer in the specified format. DO NOT provide option rationale.

---

Figure 16: Prompts for interval check and distractor generation.

