# OpenReview forum: "JointAVBench: A Benchmark for Joint Audio-Visual Reasoning Evaluation"
_ICLR.cc/2026/Conference — ICLR 2026 Poster_

### Official Review · Reviewer_EKth · 2025-10-24

**Soundness:** 2
**Presentation:** 2
**Contribution:** 2
**Rating:** 2
**Confidence:** 4

**Summary:**

This paper proposes yet another audio-visual understanding benchmark called JointAVBench. The benchmark is trying to capture audio-visual correlation in videos, especially focusing on questions that cannot be answered by visual information alone. The authors also mention the idea about multi scene-spans and emphasize that their benchmark exhibits multiple scenes.

**Strengths:**

1. Paying attention to the role of audio in video understanding is needed in the research community.

2. They provided webpage for the benchmark and their experiments spans a range of audio-visual LLMs, which might be useful information for the research community.

**Weaknesses:**

1. Technical Contribution:

I have significant doubt on the technical contribution of this paper/benchmark. Given that it is mainly focused on audio in video, a clear discussion of how important the audio modality in these tasks is needed with supporting evidence in experiments. Please also compare to the following papers:

[1] Yang et al. "Audio-centric Video Understanding Benchmark without Text Shortcut", https://arxiv.org/pdf/2503.19951.
[2] Hong et al. "WorldSense: Evaluating Real-world Omnimodal Understanding for Multimodal LLMs", https://arxiv.org/abs/2502.04326

which also introduce audio-visual benchmark with audio-centric data. In particular, [1] also includes mainly questions that cannot be answered with visual modality alone, and also uses both speech and audio events, so the first two points are already being done previously unfortunately.

2. Ambiguity in the definition of __scene__:

Regarding the 3rd contribution, I am unsure why datasets like video-MME are not multi-scene. I don't see a clear difference between the proposed dataset and videos in video-MME.

3. Careless Writing:

Please correct the question marks on page 3 for those citations.

**Questions:**

See weaknesses about the definition of scenes

---

> ### Author Response · Authors · 2025-11-26
> **Response 1/2**
>
> **Q1: Concerns regarding technical contribution, the importance of the audio modality, and comparison with recent works [1, 2].**
>
> **A1**:  We thank the reviewer for their insightful and constructive questions. Beyond perception and alignment capabilities, we believe that future omni-models should also comprehend temporal and spatial contexts, character relationships, motivations, and complex narratives by leveraging both visual and auditory signals. Films serve as an ideal testbed for these capabilities, offering challenging scenarios complementary to real-world audiovisual contexts.
>
> We construct JointAVBench for the following reasons:
> 1. **To avoid data leakage.** Most existing movie datasets are built upon popular long films, which often have extensive textual summaries and discussions available online—materials likely used in the training of large language models (LLMs). This makes it difficult to distinguish multimodal reasoning from knowledge recall. To avoid this uncertainty, we chose a new dataset consisting of less popular short films to build our benchmark.
> 2. **To enforce multimodal reasoning.** Many current datasets contain questions answerable using visual information alone, failing to adequately assess multimodal reasoning or to explore how models balance different modalities.
> 3. **To increase challenge.** With leading omni-models already achieving high performance on benchmarks such as Video-MME (e.g., video-SALMONN 2+ achieving more than 70% accuracy in Video-MME [1]), we believe the community requires more challenging tasks to inspire further innovation.
>
> Accordingly, we introduce JointAVBench, which differs from existing benchmarks such as AVUT and WorldSense in the following key aspects:
>
> 1. Comparison with AVUT [1]: AVUT focuses primarily on audio-visual alignment and audio understanding, whereas JointAVBench targets higher-level joint reasoning abilities. We define **15 fine-grained reasoning tasks, in contrast to AVUT’s 3 alignment tasks** (with 2 additional open-ended tasks yet to be released). Moreover, our benchmark incorporates **four types of audio** (speech, sound events, music, and vocal traits), while **AVUT covers only two** (speech and sound events).
> 2. Comparison with WorldSense [2]: First, as shown in Table R1, **only 62.9% of WorldSense questions require joint audio-visual reasoning**, meaning a significant portion can be solved with unimodal shortcuts. In contrast, **all questions in JointAVBench are designed to be unsolvable using only visual or audio input**, ensuring an evaluation of multimodal fusion. Second, JointAVBench **includes vocal traits** — an important dimension of human communication seldom addressed in prior work. This type of audio signal contains abundant information about human emotion, gender, and age, and therefore is vital in understanding videos.
>
> Table R1. Comparison between these related works.  “*” indicates that we only test the ratio on audio-visual tasks of AVUT.
>
> |  | Num. of defined audio-visual tasks | Num. of required audio signal types | AV Correlated Question Ratio |
> | --- | --- | --- | --- |
> | AVUT | 5 | 2 | 77.8%* |
> | WorldSense | 26 | 3 | 62.9% |
> | JointAVBench (Ours) | 15 | 4 | 100% |
>
> We conducted an experiment on how important the audio modality is in these tasks as evidence, as described in Section 4.2 of our main paper. We first define $N_o$ (Number of Outperforms) as the count of tasks where the A+V performance is strictly better than both A-only and V-only performance, and $N_u$ (Number of Underperforms) as the count of tasks where A+V is worse than both single-modality baselines. Then, as shown in Table R2, for all evaluated models, $N_o$ **significantly outweighs** $N_u$, confirming that integrating audio and video is fundamental to solving the tasks in JointAVBench. More importantly, as model capability improves (e.g., from VideoLLaMA2 to Qwen2.5-Omni), the ratio of $N_o$ to $N_u$ increases. This pattern strongly indicates that **stronger models are better at modality fusion,** and our benchmark effectively measures this critical joint-reasoning ability, which cannot be achieved with a single modality. Our findings consistently show that for all tested Omni-LLMs, the performance with joint audio-visual input (A+V) is fundamentally superior to using either modality alone, which emphasizes the importance of audio modality.
>
> **Table R2.** $N_{o}, N_{u}$ numbers of various models.
>
> | Model | $N_{o}$ | $N_{u}$ |
> | --- | --- | --- |
> | Qwen2.5-Omni | 8 | 1 |
> | VideoLLaMA2 | 6 | 3 |
> | OneLLM | 8 | 3 |
> | video-SALMONN | 5 | 4 |
>
> [1] Changli Tang, Yixuan Li, Yudong Yang, Jimin Zhuang, Guangzhi Sun, Wei Li, Zejun Ma, and Chao Zhang. video-salmonn 2: Captioning-enhanced audio-visual large language models. arXiv preprint arXiv:2506.15220, 2025a.

---

> ### Author Response · Authors · 2025-11-26
> **Response 2/2**
>
> **Q2: Ambiguity in the definition of "scene" and its contribution.**
>
> **A2**: We thank the reviewer for pointing out the ambiguity in our definition of 'scene'.  We originally aimed to evaluate different abilities of models by explicitly distinguishing **single-scene, cross-scene, and full-scene questions:**
>
> - **Single-scene tasks** evaluate local, intra-scene understanding.
> - **Cross-scene tasks** (e.g., Multi-plot Ordering, Plot Temporal Grounding) are specifically designed to test a model's ability to connect and reason about information from temporally distant and semantically distinct scenes, a critical and challenging capability.
> - **Full-scene tasks** assess global, holistic understanding of the entire video.
>
> By revisiting the datasets like Video-MME, we find that they also consist of some questions about long video clips, e.g., with a duration of more than 1 minute and with more than one shot. Thus, the questions are somehow cross-scene or full-scene, although they did not give explicit class labels to the questions. Our initial presentation may cause confusion and overstate its importance as a distinguishing feature of the source data. Therefore, we have **removed the 'Diverse Scenes' column from our main comparison table (Table 1) and adjusted all related claims in the revised manuscript.**
>
> **Q3: Careless Writing.**
>
> **A3:** We sincerely apologize for the careless writing errors and thank the reviewer for their keen observation. We have corrected the citation issues on page 3 and have performed a thorough proofread of the entire manuscript to fix any remaining typos and improve overall presentation.

---

> > ### Comment · Reviewer_EKth · 2025-11-28
> > **Response**
> >
> > I acknowledge the author's effort in explaining those, and I am willing to increase my score to 4 to reflect this. However, I still feel the contribution is limited since it still did not go beyond the scope of AVUT or VideoMME (i.e. what they evaluate can also be largely evaluated using existing benchmark). Therefore, it is hard for me to support this paper for acceptance.

---

> > > ### Author Response · Authors · 2025-12-04
> > >
> > > We sincerely thank the reviewer for their re-evaluation and for providing this opportunity for further clarification. We would like to provide a more detailed comparison and concrete examples to demonstrate the contributions of our work.
> > >
> > > Our primary contribution is a benchmark designed to specifically and rigorously test higher-level audio-visual joint reasoning capability, where a model must integrate and reason over interconnected audio and visual cues to arrive at an answer. This presents a distinct challenge compared to previous benchmarks that test unimodal understanding or simple cross-modal alignment.
> > >
> > > **1. Task-Level Comparison with Existing Benchmarks**
> > >
> > > To substantiate our claim, we first clarify the distinct evaluation scopes:
> > > - **Video-MME** is an important benchmark for general video understanding. However, its focus is not on audio-visual interaction. Its questions are not designed to be answered through both auditory and visual clues.
> > > - **AVUT** is more closely related, but its primary focus is on audio-centric understanding and fundamental audio-visual alignment, rather than complex, integrated reasoning.
> > >
> > > A direct task-to-task comparison in Table R1 reveals the minimal overlap between our benchmark and AVUT. Out of our 15 distinct reasoning tasks, only two have conceptual parallels with AVUT's tasks. Video-MME is not compared due to its tasks focusing mainly on low-level perception and lacking audio-visual comprehension evaluation.
> > >
> > > Crucially, **JointAVBench introduces 13 novel task categories not present in AVUT**, covering cognitive dimensions like temporal, plot, and long-form reasoning. These tasks demand a deeper synthesis of multimodal information. Furthermore, AVUT's evaluation scope is currently limited, as two of its five audio-visual tasks are open-ended and have not yet been released. Therefore, the assertion that the capabilities we measure can be "largely evaluated" by existing benchmarks is not supported by a direct comparison of their defined tasks.
> > >
> > > **Table R1: Evaluation Scope Comparison between JointAVBench and AVUT**
> > >
> > > | JointAVBench Task Name | AVUT Audio-visual Task Name |
> > > | --- | --- |
> > > | Speech-based Timepoint Localization | - |
> > > | Vision-Speech Sequence Recognition | - |
> > > | Speaker Spatial Localization | Audio-Visual Character Matching |
> > > | Sounding Object Grounding | Audio-Visual Object Matching |
> > > | Sound Event Recognition | - |
> > > | Speaker Emotion Recognition | - |
> > > | Musical Tone Inference | - |
> > > | Cross-scene Association | - |
> > > | Multi-plot Ordering | - |
> > > | Plot Development Prediction | - |
> > > | Audio Function Analysis | - |
> > > | Plot Temporal Grounding | - |
> > > | Audio-Visual Detail Memory | - |
> > > | Musical Emotion Shift Inference | - |
> > > | Character Relationship Inference | - |
> > > | - | Audio-Visual Text Matching |
> > > | - | Audio-Visual Segment Matching |
> > > | - | Audio-Visual Speaker Diarization |
> > >
> > > **2. Comparative Examples Across Various Benchmarks**
> > >
> > > To further illustrate the fundamental differences in what these benchmarks evaluate, we present a comparative analysis of representative examples.
> > >
> > > **(a) JointAVBench Example (from Figure 1 in our paper)**
> > >
> > > - **Question:** What's the emotion of the speaker who wears a brown leather jacket? (Options: Confident, Angry, Calm, Fearful)
> > > - **Analysis:** Requires **joint audio-visual reasoning**. The video contains two speakers with no obvious facial expressions. The model must first visually identify the character based on the "brown leather jacket" and then link this visual entity to his speech characteristics (e.g., a "fearful tone") from the audio to correctly infer the emotion. Neither modality alone is sufficient.
> > >
> > > **(b) WorldSense Example**
> > >
> > > - **Question:** What is the position of the metal roller door relative to the woman wearing white in the video?
> > > - **Analysis:** A **visual-only task**. The answer is determined by analyzing the spatial arrangement of objects within the video frames. The audio track (a music performance) is irrelevant to answering the question.
> > >
> > > **(c) AVUT Example**
> > >
> > > - **Question:** At what point in the video did the woman mention this pop-out tool first?
> > > - **Analysis:** An **audio-only task**. The video contains only a woman’s speech, and the answer is found by locating the specific spoken phrase "pop out tool" within the audio transcript and identifying its corresponding timestamp. Visual information is not required.
> > >
> > > **(d) Video-MME Example**
> > >
> > > - **Question:** How many red socks are above the fireplace at the end of this video?
> > > - **Analysis:** A **visual-only task**. The model needs to count specific objects in the video. The audio, which provides background information on Christmas traditions, does not contribute to the answer.
> > >
> > > These examples clearly show that while other benchmarks often permit unimodal shortcuts, every question in JointAVBench is meticulously designed to preclude such solutions, thereby forcing an evaluation of a model's **true multimodal integration and reasoning capabilities**.

---

### Official Review · Reviewer_77mz · 2025-10-26

**Soundness:** 2
**Presentation:** 3
**Contribution:** 2
**Rating:** 6
**Confidence:** 3

**Summary:**

The authors introduce a benchmark for evaluating models that jointly process audio and vision. The dataset consists of 2.8k QA pairs organized into fine-grained categories. Unlike existing benchmarks, it explicitly includes diverse audio types—speech, music, vocal traits, and sound events. In addition, the data are annotated along cognitive dimensions (e.g., temporal, spatial, emotional) and by scene type (single, multiple, or full, indicating how much of the video is needed to answer a question). The evaluation reveals that models struggle disproportionately on certain question types, and even the strongest model assessed achieves only 56% accuracy on average, highlighting substantial room for improvement.

**Strengths:**

- The dataset is automatically generated and then manually inspected, an important step that strengthens its reliability as a benchmark.
- I appreciate the fine-grained categorization of question types, which enables more detailed error analysis of current models.
- The analysis in Table 4, comparing joint modality performance with single-modality performance, is particularly insightful and underscores an important dimension for evaluating “omni-models.”

**Weaknesses:**

- Although the dataset’s fine-grained design is valuable, the effective size of each subset is necessarily small given the overall scale of 2.8k examples, which limits the strength of conclusions about model performance on specific question categories.
- While the dataset has been manually inspected, reporting human performance from an independent set of annotators would provide a useful reference point for comparison with model capabilities.

**Questions:**

Could you provide a breakdown of each question type's size in the benchmark? (as broken down in Table 2)

Typos
- line 481: "JointAVBench is" - missing space.

---

> ### Author Response · Authors · 2025-11-26
>
> **Q1: The data size of each category is relatively small, which may limit the strength of conclusions.**
>
> **A1**: We appreciate the reviewer for this thoughtful comment on the trade-off between task granularity and sample size. We made a conscious decision to prioritize diversity and quality, based on the following rationale:
>
> 1. Our benchmark construction pipeline incorporates a crucial manual inspection phase to ensure the correctness and quality of every QA pair, which is labor-intensive. For example, we discarded 28.2% of automatically generated data that did not meet our quality standards. While this limits the sheer scale, it significantly enhances the benchmark's reliability, which we believe is paramount for rigorous evaluation. Nevertheless, we fully acknowledge that expanding the dataset would further enhance its evaluation capabilities. Therefore, we plan to continuously expand the benchmark in future work.
> 2. While the absolute number of examples per category is modest, it is comparable to or even larger than other reasoning benchmarks. For example, as shown in Table R1, our benchmark provides a substantially larger number of samples per reasoning task compared to recent works like WorldSense. This ensures that our per-category conclusions are based on a reasonable number of data points. A detailed distribution of tasks can be found in Figure 8(e).
>
> Table R1. Comparison of the proportion of task size. |Task| denotes the number of samples of this task.
>
> |  | &#124;Task&#124; ≤ 100 | 100 < &#124;Task&#124; ≤ 200 | &#124;Task&#124; > 200 |
> | :-: | :-: | :-: | :-: |
> | WorldSense | 38.5% | 57.7% | 3.8% |
> | JointAVBench | 6.7% | 53.3% | 40.0% |
>
> **Q2: Lack of human performance as a reference point.**
>
> **A2**: We thank the reviewer for this excellent suggestion. In response to this valuable feedback, we have already initiated a formal human performance evaluation. According to the 400 human evaluation samples we have collected, **the average performance of humans is 88.1%**. We observe that there’s a huge gap between humans and the best omni-LLMs (25.5% accuracy gap), leaving enough space for Omni-LLMs to improve. We are actively collecting the results and commit to including the full, detailed analysis in the paper.
>
> **Q3: Breakdown of each question type’s size.**
>
> **A3:** We thank the reviewer for this request. We also value this question, and we have discussed the reasons that our task size isn’t uniformly distributed in Q1. We updated and added the absolute number of different tasks in Figure 8(a), which indicates that the majority of our tasks are relatively large in size. We have also updated a comparison of our tasks’ distribution with the related WorldSense dataset in Figure 8(e), which illustrates that the size of our sub-tasks is quite large and enough to evaluate models. We will continue to add more samples to enrich our benchmark in the future.
>
> **Q4: Typos**
>
> **A4:** We sincerely thank the reviewer for carefully reading our manuscript and pointing out the typo. We have corrected the specified error on line 481 and have conducted another thorough proofread of the entire paper to correct any other remaining typographical errors.

---

> > ### Comment · Reviewer_77mz · 2025-11-27
> >
> > Thank you for the response. I will maintain my current positive score.

---

### Official Review · Reviewer_Qjdf · 2025-11-01

**Soundness:** 3
**Presentation:** 3
**Contribution:** 3
**Rating:** 6
**Confidence:** 4

**Summary:**

This paper introduces JointAVBench, a new benchmark designed to evaluate the joint audio-visual reasoning capabilities of Omni-LLMs. JointAVBench is organized according to a comprehensive, multi-dimensional taxonomy. With the proposed three-stage semi-automated generation pipeline, the data acceptance rate during human verification reached 71.8%, significantly improving data collection efficiency. An extensive evaluation of current Omni-LLMs, Video-LLMs, and Audio-LLMs on JointAVBench highlights significant performance gaps and specific areas of weakness.

**Strengths:**

- The paper is well-written and easy to follow. The proposed semi-automated benchmark generation pipeline is reasonable and and effectively lowers data labeling costs.
- The comparative analysis of various models reveals the limitations of existing models, which in turn reflects the value of the benchmark.

**Weaknesses:**

The benchmark's exclusive reliance on pre-existing datasets raises two methodological concerns. Firstly, it compromises the integrity of the evaluation, as the test data may be contaminated (i.e., previously seen by models) or decontextualized. Secondly, this approach circumvents the foundational challenge of raw data acquisition and curation. The fidelity of any benchmark is fundamentally contingent on its source data, and a robust construction pipeline should therefore incorporate an automated or semi-automated mechanism for this essential process.

**Questions:**

Given the submission deadline, the exclusion of some recent models from the evaluation on JointAVBench is understandable. Nevertheless, the paper would be significantly strengthened if the authors could include performance metrics for models like Qwen3-Omni, video-SALMONN-2+, and Gemini-2.5-pro, as this would provide a more comprehensive reflection of the current capabilities of Omni-LLMs.

---

> ### Author Response · Authors · 2025-11-26
>
> **Q1: Data may be contaminated.**
>
> **A1**: We sincerely thank the reviewer for raising this critical point. Data contamination was a primary consideration during the design of our benchmark. We have taken several measures to mitigate this risk:
>
> 1. **Data Sourcing from a Specialized Domain:** To minimize the probability of our test data appearing in large-scale training corpora, we deliberately chose independent short films to be the source. Unlike popular movies or widely used video datasets, these films are less likely to have been scraped and included in LLMs’ training data. Our selected datasets also exclusively feature recently released films, with the majority of films released after 2021, which further reduces the chance for potential contamination.
> 2. **Novelty in Task Formulation:** Even if a model has been exposed to the source videos, our benchmark evaluates its ability to answer novel, complex questions that require joint audio-visual reasoning ability. We use human-designed question templates to generate these questions, ensuring they are not simple factual recalls but demand deeper comprehension. This moves the challenge from recognition to reasoning.
>
> Besides, our evaluation results in Table 3 further provide strong evidence that the benchmark remains challenging for even the most advanced models. As shown in Table R1, state-of-the-art Omni-LLMs still struggle with many tasks in JointAVBench. For example, Gemini 2.5 Pro achieves only 35.2% on SPER (Speaker Emotion Recognition), and Qwen3-Omni scores 30.9% on PTG (Plot Temporal Grounding). This unsaturated performance strongly suggests that our benchmark effectively tests their reasoning capabilities rather than their knowledge base.
>
> **Q2: Evaluation of recent models.**
>
> **A2**: We thank the reviewer for this constructive suggestion. Providing a comprehensive reflection of the current capabilities of Omni-LLMs is crucial. In response, we have benchmarked several recently released state-of-the-art models, including Gemini 2.5 Pro, Qwen3-Omni, and video-SALMONN-2+. We have updated our paper with these results and expanded our analysis.
>
> The new evaluation results are presented in Table R1. Based on these updated results, we have derived several new insights:
>
> 1. **Performance Gap in Temporal Reasoning.** The new results reveal a significant performance gap between proprietary models and open-source models, particularly in temporal tasks. For instance, Gemini 2.5 Pro outperforms the best open-source model (Qwen3-Omni) by over 20% in MPO (Multi-plot Ordering) and nearly 30% in PTG (Plot Temporal Grounding). This suggests that advanced temporal understanding and grounding of audio-visual events remain a key challenge for current open-source Omni-LLMs.
> 2. **Underperform in vocal traits tasks.** We observe that though new evaluation results provide better results, tasks that require vocal traits remain underperformed. For example, though Gemini2.5-pro pushes SPL (Speaker Spatial Localization) to nearly 60% accuracy, it still underperforms in comparison to other tasks. This observation suggests that future research should focus not only on speech text, but also on the auditory information beyond text.
>
> Table R1. Performance of recently released models.
>
> | Model Name | Parameter Size | Avg | STL | SPL | SOOG | SOER | SPER | MPTI | VSSR | CSA | MPO | PTG | AFA | PDP | AVDM | MESI | CRI |
> | --- | --- | --- | --- | --- | --- | --- | --- | --- | --- | --- | --- | --- | --- | --- | --- | --- | --- |
> | Gemini2.5Pro | - | 62.6% | 73.0% | 59.4% | 60.8% | 68.9% | 35.2% | 68.1% | 76.5% | 43.8% | 66.0% | 60.7% | 65.5% | 45.7% | 75.5% | 66.1% | 81.9% |
> | Qwen3Omni | 30B | 62.1% | 71.1% | 43.4% | 73.8% | 78.4% | 35.7% | 80.3% | 75.7% | 42.1% | 45.2% | 30.9% | 59.7% | 47.3% | 61.8% | 69.2% | 84.0% |
> | video-SALMONN2+ | 7B | 47.3% | 40.9% | 23.2% | 58.4% | 59.1% | 17.6% | 72.8% | 52.6% | 30.8% | 40.4% | 22.1% | 55.7% | 45.2% | 39.0% | 70.7% | 61.6% |

---

### Official Review · Reviewer_1yxd · 2025-11-01

**Soundness:** 2
**Presentation:** 3
**Contribution:** 2
**Rating:** 4
**Confidence:** 4

**Summary:**

The JointAVBench benchmark proposed a comprehensive evaluation tool that covers 5 cognitive dimensions (temporal, spatial, emotional, plot, and long-form), 4 audio types (speech, sound events, music, and vocal traits), and 3 scene spans (single-scene, cross-scene, and full-scene). It addresses the shortcomings of existing datasets in terms of multimodal dependency, audio diversity, and scene complexity.

**Strengths:**

- Achieves a 100% Audio-Visual Correlation Ratio (AV Corr. Ratio), where every question requires the integration of both audio and visual information to answer, avoiding the flaw that single-modality can solve questions in existing benchmarks.
- Conducts a comprehensive evaluation of three types of mainstream models (Omni-LLMs, Video-LLMs, Audio-LLMs) and analyzes performance differences across audio types, scene spans, and cognitive dimensions, providing in-depth insights for model optimization.

**Weaknesses:**

- The benchmark exclusively uses the SF20K short-film dataset (1,046 films after filtering) as its video source, but this choice introduces severe scene bias and data homogeneity: Films are professionally produced, narrative-driven content—they do not reflect real-world audio-visual scenarios (e.g., surveillance footage, live streams, industrial monitoring, daily vlogs) where Omni-LLMs are likely to be applied.
- The "semi-automated pipeline" (omni-caption generation → QA creation → quality control) relies heavily on LLMs (Qwen2.5-Omni, Qwen2.5-VL) but fails to validate the accuracy and absence of hallucinations in LLM outputs—this undermines the entire benchmark’s credibility.
- Meanwhile, the authors use Qwen2.5-Omni and Qwen2.5-VL, and having these models serve as both annotation models and evaluation models will introduce bias. This causes the answers to be more inclined towards the understanding and responses of these two models, leading to biased evaluation.
- The dataset adopts movies, which focus heavily on verbal dialogue. This results in greater bias in the evaluation of natural sounds and music, making the dataset unsuitable to be used as a benchmark for evaluation.
- No information on annotator qualifications (e.g., whether they have expertise in audio-visual analysis) or inter-annotator agreement (e.g., Cohen’s Kappa coefficient). If two annotators disagree on 30% of QAs, the "accepted" data is subjective and unreliable.
- The manuscript uses "official codebase with default configurations" for open-source models  but "official APIs" for closed-source models . However, it does not disclose API parameters (e.g., max tokens) or whether closed-source models used larger parameter sizes  This makes performance comparisons meaningless—higher accuracy could stem from larger model size, not better joint reasoning.

**Questions:**

See weakness.

---

> ### Author Response · Authors · 2025-11-26
> **Response 1/2**
>
> We sincerely thank the reviewer for their thorough review and constructive feedback. These insightful comments have helped us to significantly improve the clarity and rigor of our work.
>
> **Q1: The benchmark's reliance on a single data source may introduce scene bias and limit its applicability to real-world scenarios.**
>
> **A1:** We thank the reviewer for this critical point regarding data diversity. Our decision to exclusively use films was to fill the gap in the current evaluation landscape, rather than to replicate existing efforts.
>
> Firstly, **our initial goal was to complement existing works.** We acknowledge the importance of real-world scenarios. Prominent benchmarks like WorldSense have made excellent progress in covering domains such as daily vlogs and egocentric videos. Therefore, based on our initial goal, we select films to complement recent works and incorporate SF20K as our data source.
>
> Secondly, **we select this data source to avoid the data leakage issue.** Compared with traditional long movies, short films are less likely to have been released and discussed on the internet. Therefore unlikely to be used as training data in today’s Omni-LLMs. This characteristic ensures low data leakage probability and increases the credibility of our benchmark.
>
> Thirdly, **we find films to be a perfect data source for audio-visual complex reasoning.** Films offer structured narratives, stable character shots, and complex relationships, essential for designing complex reasoning tasks. For example, tasks like Multi-plot Ordering (MPO) requires the model to understand plot developments and causality across non-adjacent scenes, which matches the multi-plot nature of films. This focus on complex reasoning is also validated by other recent works like Video-Holmes[1], which also leverage films for similar reasons.
>
> [1] Junhao Cheng, Yuying Ge, Teng Wang, Yixiao Ge, Jing Liao, and Ying Shan. Video-holmes:
> Can mllm think like holmes for complex video reasoning? arXiv preprint arXiv:2505.21374,
> 2025.
>
> **Q2&Q3: Lack of validation for LLM hallucinations and bias from using Qwen models for both annotation and evaluation.**
>
> **A2:** We sincerely appreciate the reviewer raising these crucial concerns about the benchmark's credibility. To conclude, we minimize hallucination and bias via a human-involved filtering process and a carefully designed generation pipeline. Specifically, the following procedures:
>
> 1. **Rigorous Filtering to Eliminate Hallucinations:** We incorporated both automatic and human labor to process data.
>     - **Automated Quality Control:** In stage 3 of our generation pipeline, a series of automated checks (modality, logic, ambiguity, etc., as detailed in Sec 3.2.3) were performed, resulting in discarding 56.4% of the initially generated QA pairs due to flaws like logical inconsistencies or description ambiguity.
>     - **Rigorous Human Verification:** The remaining data then underwent manual verification by human annotators. In this phase, a further 28.2% of the data was discarded for failing our quality criteria (e.g., information correctness, modality usage).
>
>     This filtering process, which **removes over 68% of the initial data**, ensures a high degree of fidelity in the final benchmark.
>
> 2. **Carefully Designed Pipeline to Minimize Model Bias:** Our pipeline is not based on one single model. We deliberately use a chain of specialized models to minimize reliance on any single model's architecture. We use Qwen2.5-VL only for visual captions, Qwen2.5-Omni only for audio captions, and Whisper-v3 only for transcription. For QA pair creation and quality control, we use a text-only Qwen-2.5 LLM guided by human-designed templates, decoupling it from the omni-modal architecture of the models being tested.
>
> As indicated in Table 3, our results show that the Qwen-series models (e.g., Qwen2.5-Omni) do not achieve near-perfect scores. In contrast, they struggle significantly on many tasks. If there were a strong inherent bias, we would expect their performance to be much higher. Besides, our updated evaluation results in Table 3 show that Gemini2.5-pro achieves the highest score, indicating that our benchmark doesn’t favor generator models.

---

> ### Author Response · Authors · 2025-11-26
> **Response 2/2**
>
> **Q4: Using movies may introduce a bias towards dialogue over natural sounds and music.**
>
> **A4:** We appreciate the reviewer’s thoughtful point. While movies are dialogue-rich, they also make extensive use of sound and music for storytelling. We quantitatively analyzed our benchmark to ensure a balanced representation.
>
> 1. **Quantitative Distribution of Audio Types:** We calculated the proportion of questions in our benchmark that rely on each of the four audio types. As shown in Table R1, sound events and music constitute a substantial portion of our benchmark, demonstrating a balanced focus beyond just speech.
> 2. **Comparison with Related Work:** We provide a comparison with WorldSense, which incorporates an open-domain data source. Our benchmark features a more realistic distribution across different audio types, e.g., speech more frequently provides useful information than music, and for the first time includes vocal traits. This ensures that models are evaluated on a wider spectrum of auditory reasoning skills.
>
> **Table R1.** Distribution of questions by required audio type.
>
> | Audio Type | **JointAVBench (Ours)** | WorldSense |
> | --- | --- | --- |
> | Speech | 30.7% | 42.4% |
> | Sound Event | 25.7% | 31.3% |
> | Music | 18.0% | 26.3% |
> | Vocal Traits | 22.8% | - |
>
> **Q5: Lack of information on annotator qualifications and inter-annotator agreement**
>
> **A5:** We thank the reviewer for highlighting this omission, which is critical for ensuring the reliability of our human verification process.
>
> 1. **Annotator Qualifications:** Our annotation team is provided by a professional crowdsourcing labeling company. Each of our annotators has a bachelor's degree from a high-ranked university, is proficient in English, and passed a qualification test on sample tasks before participating.
> 2. **Inter-Annotator Agreement:** We agree that reporting Inter-Annotator Agreement (IAA) is crucial. To assess the reliability of our annotations, we randomly selected 400 samples and had them independently annotated by two annotators. We then calculated Cohen's Kappa coefficient to measure the agreement. The resulting **Cohen's Kappa was 0.713, which is interpreted as "Substantial Agreement"** according to established benchmarks [1]. This strong level of agreement validates the clarity of our annotation guidelines and the reliability of our final dataset. We have added these details to the Appendix of our manuscript.
>
> [1] J Richard Landis and Gary G Koch. The measurement of observer agreement for categorical data. biometrics, pp. 159–174, 1977.
>
> **Q6: Lack of details on closed-source API configurations, hindering fair comparison.**
>
> **A6**: Good remarks! We have now included the parameter size in Table 3 in the revised manuscript and detailed the exact experimental settings for all models.
>
> - **Open-Source Models:** We confirm that all open-source models are evaluated in their ~7B parameter versions (with the exception of InternVL2.5-8B and Qwen3-Omni-30B) using their official codebases and a unified setting of 32 sampled frames.
> - **Closed-Source Models:** For models like Gemini 2.5 Pro, we used their official public APIs with default settings for parameters such as temperature =1.0, top_p = 0.95, top_k = 64 and output_token_limit = 65536. We sincerely appreciate the reviewer for mentioning this point, and we have added relevant details in the Appendix.

---

### Meta-Review · Area_Chair_r1Nr · 2026-01-11

**Summary:**

The paper introduces JointAVBench, aiming to evaluate the joint audio-visual capabilities of Omni-LLMs. The creation of this benchmark is automated (by leveraging advanced models), generating questions that strictly require both modalities to answer.

The original ratings were mixed. Overall, the reviewers appreciated the fine-grained taxonomy, the inclusion of diverse audio types, and the rigor of the proposed generation pipeline involving manual verification. The initial major concerns are: 1) the overlap with existing benchmarks; 2) the exclusive reliance on movie data (e.g., may cause domain bias and data contamination); 3) the reliability of annotators is not provided/analyzed; and 4) more comprehensive baselines (e.g., human performance, newer models) should be added.

The rebuttal responded well to these concerns. As detailed in the next part, the AC believes most of the major concerns are well addressed. The remaining one is that this benchmark is built exclusively with SF20K short-film dataset, which could limit its applicability to "real-world" scenarios. However, the AC believes this is not significant as the analysis here is still research meaningful and the strategies could be leveraged to other data sources for more comprehensive benchmark creation. But the authors should clearly discuss this limitation in the final version to provide future readers with a better context. Therefore, the final decision is acceptance.

**Reviewer Concerns:**

Addressed concerns:
1) Overlap with existing benchmarks: In the rebuttal, the authors provide detailed task comparisons, which sufficiently differentiate their work from prior benchmarks.
2) Reliability of human annotators: The authors provided details on the professional team and a Cohen’s Kappa score of 0.713, suggesting a reliable agreement.
3) More baselines: In the rebuttal, the performance of multiple advanced models like Gemini-2.5 and human is provided.

Outstanding concerns:
1) Domain bias: As this dataset is built exclusively with SF20K short-film dataset, the domain bias is expected, and its applicability to real-world scenarios will be limited.

**Reviewer Scores:**

It is expected that Reviewer 1yxd would increase the score to 6, as most of the raised concerns are well addressed (except for domain bias).

It is expected that Reviewer Qjdf and Reviewer 77mz would maintain their positive scores.

It is expected that Reviewer EKth still maintains a slightly negative score (also partially confirmed in the response); but the AC does not see a significant weakness in this paper's contribution (and believes its contribution is sufficiently different from prior works).

---

### Decision · Program_Chairs · 2026-01-26

Accept (Poster)